# InSAR sensitivity to active layer ground ice content in Adventdalen, Svalbard

Lotte Wendt[1,2,3], Line Rouyet[2], Hanne H. Christiansen[3], Tom Rune Lauknes[2], Sebastian Westermann[1,4]

[1] Department of Geosciences, University of Oslo, P.O. Box 1047 Blindern, 0316, Oslo, Norway
[2] NORCE Norwegian Research Centre AS, Siva Innovasjonssenter, P.O. Box 6434, 9294, Tromsø, Norway
[3] Arctic Geophysics Department, The University Centre in Svalbard (UNIS), P.O. Box 156, 9171, Longyearbyen, Norway
[4] Centre for Biogeochemistry in the Anthropocene, University of Oslo, Oslo, Norway

*Correspondence to*: Lotte Wendt (lottew@uio.no)

## Abstract

Interferometric Synthetic Aperture Radar (InSAR) remote sensing of surface displacement in permafrost environments has the potential to resolve ground ice dynamics and potentially active layer thickness, yet field validation is sparse. Here we present a comparison between in-situ ground ice contents and the seasonal InSAR displacements of the following thawing season at 12 coring sites in Adventdalen, Svalbard. The study is focused on the year 2023, where frozen sediment cores were collected at the end of spring from the active layer and the uppermost permafrost. The sediment cores were analyzed with high resolution

for volumetric ground ice and excess ice contents. The active layer thickness was estimated by probing the thaw depth at the end of the thawing season 2023, allowing estimation of the amount of expected subsidence from seasonal ground ice melt. The InSAR vertical displacements for the thawing season were derived from Small Baseline Subset (SBAS) processing of Sentinel-1 imagery. The expected subsidence from ground ice melt within the measured active layer aligned well with the seasonal InSAR maximum vertical displacement. Monte Carlo simulations were performed to include uncertainties in the expected and

measured InSAR subsidence, leading to a mean coefficient of determination of 0.68 and a mean absolute error of 15 mm for the correlation between InSAR subsidence and expected subsidence from in-situ ground ice melt. Excess ice is highly variable and is the main source of the expected subsidence during this thawing season, which was exceptionally warm. The expected subsidence and active layer thickness show only a weak relationship due to the observed complex ice content distribution in the active layer and uppermost permafrost. Our results show the significant potential of InSAR for mapping ground ice

variability; however, they also suggest that estimating active layer thickness using InSAR requires careful consideration of the complex occurrence of both pore and excess ice in the active layer and uppermost permafrost.

## 1 Introduction

Permafrost environments underlie approximately 15% of the northern hemisphere and are highly sensitive to ongoing climate change (Obu, 2021, Biskaborn et al., 2019). Overlying the permafrost is the active layer, which thaws during summer and

refreezes during winter. An increase in the active layer thickness (ALT) serves as a key indicator of permafrost degradation

(GCOS, 2022). The formation and melt of ground ice impacts the thermal behavior of the ground, with larger ground ice contents increasing the latent heat of fusion for thaw and thus causing a reduced seasonal thaw depth (French, 2007a). Seasonal variations in the active layer ground ice content affect local hydrology and ground stability (Walvoord and Kurylyk, 2016). Long-term ground ice loss is associated with active layer deepening and may cause pronounced terrain alternations,

such as subsidence and the development of thermokarst landforms (Burn et al., 2024). These changes have broad implications for the carbon cycle in permafrost areas (Schuur et al., 2015).

Traditional methods for monitoring ALT and mapping ground ice (e.g. thaw depth probing, temperature monitoring in boreholes, drilling and geomorphological surveys, thaw tubes) typically rely on labor-intensive, time-consuming in-situ surveys. Additionally, many regions are difficult to access for in-situ monitoring (Bonnaventure and Lamoureux, 2013).

Interferometric Synthetic Aperture Radar (InSAR) satellite remote sensing has in recent years been increasingly used to monitor thaw subsidence in permafrost environments. Studies have shown that the spatiotemporal InSAR variability can be related to ground ice content patterns and soil water contents (Daout et al., 2017; Chen et al., 2020), as well as air and ground temperature variations (Strozzi et al., 2018; Bartsch et al., 2019), and different landforms and surface materials (Rouyet et al. 2019). The detected InSAR thaw subsidence has been exploited to inversely retrieve ALT assuming certain ground conditions

(e.g. Liu et al., 2012; Schaefer et al., 2015). Further, InSAR subsidence measurements have proven effective for monitoring long-term ground ice loss due to permafrost degradation (e.g. Wang et al., 2023). InSAR allows large-scale mapping of surface displacements related to active layer and permafrost changes, independent of solar illumination and cloud conditions. However, the detailed response of InSAR measurements to seasonal ground ice dynamics remains underexplored.

Ground ice, which varies in distribution, includes both pore ice within soil pores and excess ice that exceeds the soil's pore

space (Everdingen, 1998). Pore ice forms when soil moisture freezes within the existing pore spaces of mineral and organic soils. Depending on the degree of saturation, the phase change causes either a volume expansion within the pore space or, if the pores are saturated with water, an expansion of the soil structure itself due to the pressure exerted by the growing ice. Upon melting, pore ice does not produce water in excess of the pore space. Ice segregation processes can cause the migration of water towards the freezing front, enriching soil pores further with ice and leading to the growth of ice lenses (Derek and Miller,

1966). If the accumulation of ice exceeds the pore space, excess ice forms, which upon melting produces water in excess of the pore space (Taber, 1930; Rempel et al., 2007).

The melting of pore ice in saturated ground can lead to thaw consolidation, caused by the volume loss associated with the density difference between ice and water (approx. 8%) (Dumais and Konrad, 2024). The melting of excess ice has an even more pronounced effect, as the loss of ice that exceeds the soil's pore space can cause significant subsidence when the resulting

meltwater drains away (Morgenstern and Nixon, 1971).

Comprehensive field validation to measure the impact these ice types have on InSAR measurements is still lacking, underscoring the need for improved understanding of how InSAR captures these seasonal ground ice changes (Bartsch et al., 2023).

The contribution of excess ice melt to observed seasonal surface subsidence was hypothesized by Liu and Larson (2018) and Bartsch et al. (2019), yet without further in-situ field validation with active layer ground ice contents. Studies by Zwieback and Meyer (2021) and Zwieback et al. (2024) indicate that InSAR subsidence measurements from the late thawing season can suggest excess ice melt from the top of permafrost. These studies highlight the need for further field validation to understand how well InSAR displacements align with seasonal ground ice melt in the active layer. The sensitivity of InSAR to pore ice and excess ice melt as well as the correlation between active layer thickness and InSAR subsidence are also of relevance for the inverse retrieval of active layer thickness from InSAR (Liu et al., 2012).

Here we present a comparison between seasonal C-band InSAR displacements and in-situ active layer and uppermost permafrost ground ice contents from different periglacial landforms in the Adventdalen valley, Svalbard. Our objectives are to (1) compare the InSAR subsidence to the expected subsidence from ground ice melt based on in-situ field measurements, (2) investigate the contributions of pore and excess ice melt to the seasonal subsidence signal, and (3) evaluate the relationship between subsidence magnitude and ALT.

## 2 Methods

### 2.1. Study area

The study is focused on the valley Adventdalen, located in Central Spitsbergen, Svalbard (78.2°N, 15.8°E) (Fig. 1). This valley has continuous permafrost, varying in thickness from less than 100 m in valley bottoms and coastal areas to up to 500 m in the mountains (Humlum et al., 2003). The uppermost permafrost formed largely syngenetically due to eolian deposition on the fluvial terraces adjacent to the river Adventelva during the late Holocene (Gilbert et al., 2018) (Fig. 1B). The periglacial valley is covered by landforms such as ice-wedge polygons, solifluction sheets, alluvial fans, loess terraces on the sides of Adventelva, and moraine deposits. The valley bottom has extensive tundra vegetation, whereas there is no vegetation in the braided river plain. The landcover is favorable for the application of InSAR with C-band sensors, since decorrelation by shrub vegetation does not occur (Wang et al., 2020).

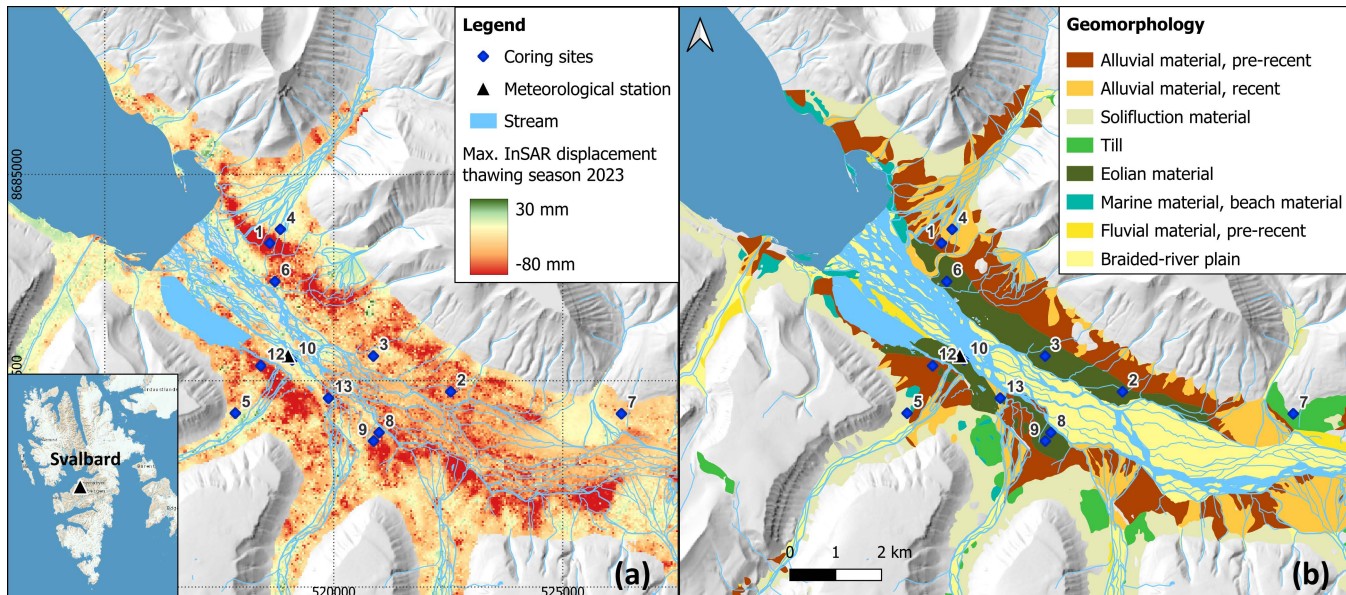

**Figure 1:** (a) The Adventdalen study area with the location of the coring sites and their label numbers. The background is the maximum InSAR seasonal displacement of 2023. Subsidence is shown with negative values (red) and heave with positive values (green). Note that the colour scale is saturated for visualization. (b) Simplified geomorphological map of the study area with the main sediment deposits (Rouyet et al., 2019; modified from Härtel and Christiansen, 2014). Background: hillshade of a digital elevation model (Norwegian Polar Institute, 2014a,b). Coordinate System: WGS 1984 UTM 33N.

The climate of the study area is classified as Arctic tundra with maritime influence according to the Köppen-Geiger classification (Kottek et al., 2006; Eckerstorfer and Christiansen, 2011). Historical data from 1971 to 2000 indicate that the mean annual air temperature (MAAT) at Svalbard Airport, located about 10 km west of Adventdalen, was -5.8 °C and the mean annual precipitation was 196 mm (Hanssen-Bauer et al., 2019). From 1971 to 2000, the MAAT increased by approximately 1 °C per decade, with the most pronounced warming during the winter months. During the same period, precipitation increased, especially in autumn and winter (Hanssen-Bauer et al., 2019). The permafrost in Svalbard has been observed to be warming, with ALT increasing by 0.6 cm/year from 2000 to 2017 at the UNISCALM grid near coring site E10 (Fig. 1A) (Hanssen-Bauer et al., 2019).

Our study focuses on the thawing seasons of 2021 and 2023. The thawing season 2023 followed the sediment core collection in spring 2023, enabling direct comparisons of InSAR subsidence and thaw progression with active layer ground ice measurements. The thawing season 2023 was exceptionally warm, similar to the previous warm summers 2020 and 2022 (Fig. 2B). Yet, 2023 was also very wet (Fig. 2B). In contrast, the thawing season 2021 was marked by exceptionally cold and dry conditions (Fig. 2B), though it lacks in-situ ground ice measurements for comparison.

The warm thawing season of 2023 spanned from the end of May to mid-September, while the cooler thawing season 2021 started in June and lasted into October (Fig. 2A). Due to the exceptionally warm conditions of summer 2023, the ALT recorded at boreholes and the Circumpolar Active Layer Monitoring (CALM) sites in Adventdalen reached unprecedented depths (Norwegian Meteorological Institute 2024a).

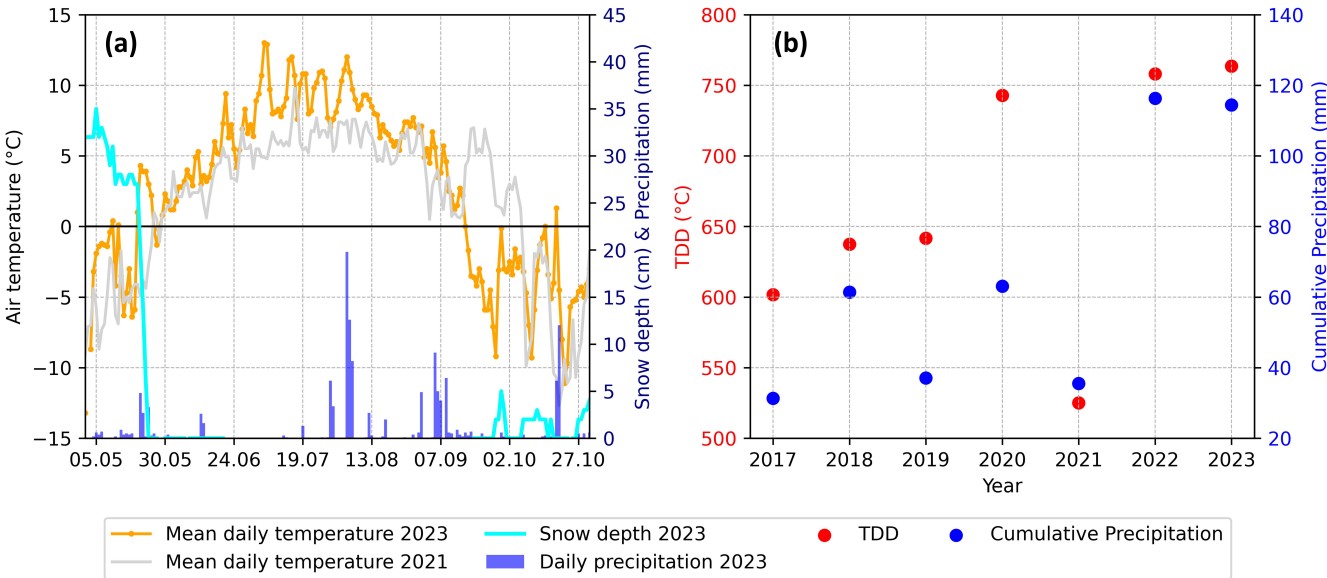

**Figure 2:** Meteorological conditions in the investigated thawing periods. (a) Air temperature for 2021 and 2023, as well as the precipitation and snow depth data from the thawing season 2023. (b) Overview of the Thawing Degree Days (TDD) and cumulative precipitation for the thawing seasons 2017 – 2023 (based on the individual start dates of thawing periods until 15 September each year). The data is from the Adventdalen meteorological station, except for the cumulative precipitation of 2021, which is from the Svalbard airport (10 km away), due to instrument malfunctioning (Norwegian Meteorological Institute 2024b).

The coring sites were selected to cover a variety of anticipated InSAR subsidence magnitudes and time series patterns as well as diverse ground conditions. Analysis of seasonal InSAR displacements from 2017 to 2022, alongside the findings of Rouyet et al. (2019), guided the selection of sites exhibiting varied InSAR displacement time series (Fig. 1A). Additionally, the choice of sites was informed by the geomorphological map of Härtel and Christiansen (2014), ensuring a comprehensive representation of different sediment deposit types in the lowland of Adventdalen (Fig. 1B). The selection process also considered various landforms, such as sampling alluvial fans at both upper and lower locations, to capture a range of sediment grain sizes. Further, soil moisture conditions were considered to include dry and wet locations based on the NDWI (Gao, 1996) remote sensing index of Sentinel-2 imagery from summer 2022. To minimize the difference between the point-scale coring result and the larger InSAR pixel size, coring was specifically conducted in areas exhibiting minimal spatial variability within the InSAR pixels surrounding the coring location. Field pictures of the 12 coring sites are shown in Fig. S1.

## 2.2. Sediment coring and in-situ measurements of thaw depth

Frozen sediment cores containing the entire active layer (depths 0.6-2.0 m, Table 1) and top permafrost were retrieved from April 15 to May 01, 2023, in winter conditions with negative air temperatures. The drilling was conducted using a STIHL™ BT 131 Earth Auger equipped with 0.5 m long core barrels. Decreasing barrel diameters were used with increasing drilling depth. The core barrel diameters were 6 cm, 5.5 cm, and 5 cm. Cores were retrieved in 5-30 cm sections and were immediately packed, air-sealed, and stored at the end of each field day in a container with an active freezing system. The samples remained

frozen in the container at the University Centre in Svalbard (UNIS), until analyzed in the laboratory. In total, 16.5 meters of sediment were cored at 12 coring sites. The coring depth ranged from 0.63 m to 2.02 m and was chosen in the field based on

the expected active layer in different landforms following previously reported ALT (Cable et al., 2018). Overall, 9.5 out of 16.5 m of cores were retrieved intact, and the remaining disturbed.

The thaw depth at each site was manually probed at the end of the thawing season in early September 2023 to determine the ALT. Measurements were taken not only at the coring locations but also 10 and 20 meters away in each cardinal direction to estimate an average ALT of the area covered by the InSAR results. However, at three sites (A1, A4, S5), the active layer was

140 only measurable at the coring location due to abundant gravel in the active layer of the surrounding area. At site T7, only 5 measurements instead of 9 could be made due to abundant rocks. Since ground ice melt leads to surface subsidence over the thawing season, the thaw depth measurements did not directly correspond to the lengths of the sediment cores, which were retrieved when the active layer was frozen (O'Neill et al., 2023). To account for this, we applied a correction based on the expected subsidence (Table 1), derived from the core data (see Section 2.4). The further used ALT was calculated as the sum

of the probed thaw depth and the subsidence correction.

**Table 1:** Overview of the coring sites, including site description, drilling date and depth, ALT date and thaw depth with standard deviation (SD) as well as subsidence correction, and the main material type and organic layer thickness. The coring sites are named after sediment deposit type and number of drilling order. The sediment deposit types are abbreviated as A (Alluvial), E (Eolian), S (Solifluction) and T (Till) and based on the geomorphological map of Härtel and Christiansen (2014).

| Coring site | Site description | Location UTM 33X; Slope angle | Drilling date; Drilling depth (m) | ALT date; Thaw depth (± SD) (m); Subsidence correction (m) | Main active layer material type; organic layer thickness (m) |
|---|---|---|---|---|---|
| A1 | Lower alluvial fan with pre-recent alluvial sediments, both fine- and coarse-grained. Dry ground conditions. | N 8683501; E 518601 2° | 16.04.2023; 2.02; | 10.09.2023; 1.41 (± na); + 0.1 | Gravel; 0.02 |
| E2 | Loess terrace with ice-wedge polygons. Drill location in centre of ice wedge polygon. Wet ground conditions. | N 8680259; E 522554 0° | 17.04.2023; 1.14; | 13.09.2023; 0.52 (± 0.03); + 0.06 | Silt; 0.3 |
| E3 | Loess terrace with no visible ice-wedge polygons. Wet ground conditions. | N 8681036; E 520864 0° | 18.04.2023; 1.22; | 10.09.2023; 0.54 (± 0.04); + 0.04 | Organic; 0.6 |
| A4 | Upper alluvial fan with recent coarse-grained alluvial sediments. Dry ground conditions. | N 8683802; E 518833 2° | 21.04.2023; 1.95; | 10.09.2023; 1.96 (± na); + 0.01 | Gravel; 0 |
| S5 | Outer solifluction sheet of Endalen. Wet ground conditions. | N 8679790; E 517849 8° | 22.04. & 24.06.2023; 1.52 | 13.09.2023; 1.37 (± na); + 0.05 | Gravel; 0.15 |
| E6 | Loess terrace with no visible ice-wedge polygons. Wet ground conditions. | N 8682663; E 518715 0° | 22.04.2023; 0.63 | 10.09.2023; 0.58 (± 0.03); + 0.05 | Silt; 0.25 |
| T7 | Moraine with fine and coarse-grained till sediments. Dry ground conditions. | N 8679778; E 526282 6° | 23.04.2023; 1.24 | 10.09.2023; 1.23 (± 0.03); + 0.04 | Gravel; 0.2 |
| E8 | Loess terrace with ice-wedge polygons. Drill location in centre of ice wedge polygon. Dry ground conditions. | N 8679367; E 520988 2° | 23.04.2023; 1.17 | 03.09.2023; 0.78 (± 0.02); + 0.06 | Silt; 0.15 |
| A9 | Outer inactive alluvial fan with pre-recent alluvial sediments. Hummocky terrain. Wet ground conditions. | N 8679183; E 520868 1° | 23.04.2023; 1.19 | 13.09.2023; 1.02 (± 0.03); + 0.08 | Silt; 0.09 |
| E10 | Elevated loess terrace. Dry ground conditions. | N 8680951; E 519092 0° | 26.04.2023; 1.37 | 03.09.2023; 1.07 (± 0.05); + 0.01 | Sand; 0.01 |
| A12 | Lower, inactive alluvial fan with pre-recent alluvial sediments. Wet ground conditions. | N 8680824; E 518407 1° | 30.04.2023; 1.22 | 13.09.2023; 0.87 (± 0.03); + 0.1 | Silt; 0.14; |
| A13 | Lower, inactive alluvial fan with pre-recent alluvial sediments. Dry ground conditions. | N 8680119; E 519884 0° | 01.05.2023; 1.37 | 13.09.2023; 0.81 (± 0.06); + 0.06 | Silt; 0.01 |

## 2.3. Analysis of ground ice contents

All intact cores were cut in half lengthwise and processed in 1–7 cm sections in a freezing lab at -8 °C. The intact subsections were scraped, the cryostratigraphy classified based on French and Shur (2010), and visual ice content and sediment type described. All intact core sections were photographed (cross-sections shown in Fig. S2), and their length $l$ and diameter $d$ measured with a ruler. Disturbed core sections were also photographed and processed in the sections retrieved during drilling without further subsampling (mean length of 9 cm ± 6 cm). For disturbed core sections, the length $l$ was measured in the field during retrieval from the borehole and the diameter $d$ was based on the core barrel diameter.

All subsamples were thawed for 12 hours in a sealed measurement beaker at room temperature. After thawing, the volume of sediments and supernatant water was measured by reading of the volumes in the graded measurement beaker with help of a ruler. The subsamples were weighed to determine their wet weight $M_w$. Afterwards, the subsamples were dried in an oven at 80 °C for 48 hours, and their dry weight $M_d$ was recorded.

The volumetric ice content (VIC) and excess ice content (EIC) were calculated based on the formulas by Kokelj and Burn (2005) and Paquette et al. (2023):

$$VIC = \frac{(M_w - M_d) * 1.09}{V_f} * 100 \qquad \text{vol [\%]} \qquad (1)$$

$$EIC = \frac{V_{sw} * 1.09}{V_f} * 100 \qquad \text{vol [\%]} \qquad (2)$$

where $V_f$ is the frozen volume of the subsample, calculated as the volume of a half cylinder from the frozen core length $l$ and diameter $d$; $M_w$ is the wet weight of the subsample (g); $M_d$ is the dry weight of the subsample (g); $V_{sw}$ is the volume of supernatant water of the thawed subsample (ml); and the factor 1.09 is to estimate the equivalent volume of ice from the water volume.

The pore ice content (PIC) was derived under the assumption that the intact core sections were saturated:

$$PIC = VIC - EIC \qquad \text{vol [\%]} \qquad (3)$$

## 2.4. Calculation of expected subsidence

The core sections were retrieved either intact or disturbed. Samples that fell apart during sampling and/or analysis (disturbed) were observed to be very dry and therefore classified as unsaturated and not considered to contribute to the subsidence signal. The intact core sections were observed to be visually saturated with pore ice and contained sometimes also excess ice based on the laboratory measurements. Based on these intact core sections, the expected subsidence was derived.

In the intact core sections, both excess ice and pore ice melt are expected to cause a volume reduction according to the density difference between ice and water (approximately 8 %) (Dumais and Konrad 2024). However, excess ice, as it exceeds the pore space of the soil column, can in addition drain or redistribute into unsaturated pore spaces, likely leading to a full volume loss of the excess ice volume (Morgenstern and Nixon, 1971). Based on this rationale, the expected subsidence $\tau$ for the thawing season 2023 was calculated from intact core sections, consisting of pore ice melt (8% volume reduction), excess ice melt (8%

volume reduction) and excess ice meltwater drainage (92% volume reduction) within the ALT measured at the end of the thawing season 2023:

$$185 \quad \tau = \sum_0^{ALT} (\underbrace{0.08 * PIC}_{\substack{melt \ of \\ pore \ ice}} + \underbrace{0.08 * EIC}_{\substack{melt \ of \\ excess \ ice}} + \underbrace{0.92 * EIC}_{\substack{drainage \ of \\ excess \ ice \ meltwater}}) * l \qquad (4)$$

Additionally, the individual contributions from the melt of pore ice, excess ice, and the drainage of excess ice meltwater to the total subsidence were calculated. The uncertainty in the expected subsidence was calculated by propagating the measurement uncertainties and is described in Appendix A.

## 2.5. InSAR surface displacement time series

The InSAR surface displacement time series were retrieved from Sentinel-1 imagery using the Small Baseline Subset (SBAS) method (Berardino et al., 2002). This technique is particularly suited for permafrost studies due to its ability to handle distributed scatterers, which is the predominant scattering mechanism on natural surfaces such as permafrost terrain (e.g. Rouyet et al., 2019; Wang et al., 2023). The SBAS algorithm utilizes a network of partially redundant, highly coherent, multi-looked interferograms to improve the signal-to-noise ratio. It separates the deformation phase from the atmospheric phase by
filtering in time and space, based on the assumption that the deformation signal is temporally correlated, whereas atmospheric effects are spatially correlated but not temporally (Berardino et al., 2002).

For summer 2023, only Sentinel-1A imagery was available, which has a revisit period of 12 days. Therefore, the minimum temporal baseline for constructing interferograms was 12 days. To mitigate temporal decorrelation and phase ambiguities from strong subsidence in the exceptionally warm summer 2023, a maximum temporal baseline of 24 days was used. This threshold
was chosen after inspection of interferograms created with longer temporal baselines, which were strongly decorrelated. All interferograms with 12- and 24-day temporal baselines were manually inspected and highly decorrelated interferograms were discarded. Both ascending and descending geometries cover the study area, but the ascending stack was incomplete due to a missing image in August 2023. After comparing time series constructed from both geometries, the descending time series was selected for further analysis due to unwrapping errors in the ascending data set. The coherence time series of all study sites
based on the further used descending data set are shown in Fig. S3.

To compare the displacements at the coring sites to a colder and drier summer, the thawing season 2021 was also processed using a descending geometry, with temporal baselines ranging from 6 to 36 days. The availability of both Sentinel-1A and 1B satellite data allowed for a shorter minimum temporal baseline whilst a longer maximum temporal baseline was possible due to higher InSAR coherence resulting from drier conditions and lower displacement rates.
Each seasonal dataset was processed independently, spanning from snowmelt to the onset of the freeze-back period. The InSAR processing was done using the GSAR software (Larsen et al., 2005), starting from Sentinel-1 Interferometric Wide (IW) swath Single Look Complex (SLC) data. Interferograms were generated using a multi-look factor of 8x2 (range x azimuth), leading to a ground resolution of approximately 18.4 meters in azimuth and 28.2 meters in range. All interferograms were filtered using an adaptive Goldstein filter (Goldstein and Werner, 1998; Baran et al., 2003). A digital elevation model

with a spatial resolution of 20 m (Norwegian Polar Institute, 2014a) was used for removing the topographic phase and for georeferencing of the results. Highly decorrelated interferograms were discarded from the analysis (see also baseline graphs in Fig. S4 and list of interferograms in Table S1). A valid pixel mask was applied based on coherence thresholds and a pixel was discarded if the coherence was below 0.42 in more than 50% of the interferograms in 2023, and below 0.5 in 2021. The interferograms were then unwrapped using the minimum-cost-flow approach of SNAPHU (Chen and Zebker, 2002), assisted by a Delauney triangulation on the sparse pixels. The unwrapping results were manually inspected for errors, but no additional interferograms were discarded. The reference point for the SBAS inversion, employing an $L_1$-norm-based cost function (Lauknes et al., 2011), was placed on the main airport building of Svalbard airport (coherence = 0.99), previously validated for InSAR analysis in this area (Rouyet et al., 2019; 2021). The turbulent atmospheric phase component was mitigated using a 500 m spatial and 24-day temporal low-pass filter.

The geocoded time series points underwent postprocessing, converting line-of-sight displacements $\varepsilon_{LOS}$ to vertical displacements $\varepsilon_{vertical}$ using the formula:

$$\varepsilon_{vertical} = \frac{\varepsilon_{LOS}}{\cos(\theta)}, \tag{5}$$

where θ is the local incidence angle in radians. Horizontal displacements were assumed negligible, given the flat terrain with a mean slope angle of 1.8° and a maximum slope angle of 8°.

To align the InSAR-derived subsidence with in-situ observations, the specific InSAR pixel covering each field site was identified. Then time series data for each coring location were extracted from the InSAR dataset for comparison. The accuracy of the InSAR subsidence was assessed by referencing studies that compared InSAR displacements to geodetic measurements in areas of mining subsidence, tectonic movement and groundwater depletion, suggesting an accuracy of approximately 5-10 mm/year (Yalvac, 2020; Cigna et al., 2021; Jiang and Lohman, 2021; Li et al., 2022). Consequently, an InSAR displacement uncertainty of ± 10 mm was assumed, although it is acknowledged that this number is a coarse estimate.

## 2.6. Statistical analysis

To assess the agreement between the expected subsidence (described in Section 2.4) and the InSAR-measured subsidence (see Section 2.5), we applied Pearson's correlation coefficient test. Recognizing inherent uncertainties in both datasets, a Monte Carlo simulation was employed. This approach involved introducing normally distributed random errors into the expected and InSAR-measured subsidence values to reflect their respective uncertainties, repeated over 1000 iterations. This process allowed us to accommodate the variability and uncertainty inherent in both measurement datasets effectively. From these iterations, average metrics were calculated, providing a more reliable estimate of the correlation between the expected subsidence and the InSAR-measured subsidence by incorporating their uncertainties.

# 3 Results

## 3.1 Variability in ALT, ground ice contents and expected subsidence

The field sites exhibited large variability in ALT, ranging from 0.6 to 2.0 m (Table 1). The ALT varied significantly ($p < 0.05$, Kruskal Wallis test) between the sites when classified by main grain size type (organics, silt, sand and gravel). The gravel sites had the largest ALT (mean 1.55 m), followed by sand (1.07 m) and silt (mean 0.83 m). The ALT was smallest at a site where the active layer was composed entirely of organics (0.58 m). At many sites, the Volumetric Ice Content (VIC) varied in the active layer, with a higher VIC in the upper active layer, followed by a decrease in VIC in the middle part of the active layer, and a slight increase in the lower active layer and uppermost permafrost (Fig. 3A). This pattern is consistent with the expected distribution of ground ice from two-sided freezing (French, 2007a). Conversely, Excess Ice Content (EIC) was generally low in the upper and central active layer and increased in the lower active layer and uppermost permafrost at most sites (Fig. 3B). The transition from layers with only pore ice to layers containing also excess ice was very sharp at several sites, meaning that there was a sudden strong increase in excess ice (see diversity in ground ice types over depth in Fig. 8, as well as Fig. S5, S6, S7). Nevertheless, not all sites showed this pattern, again displaying the large variability found in ground properties due to different geomorphological conditions.

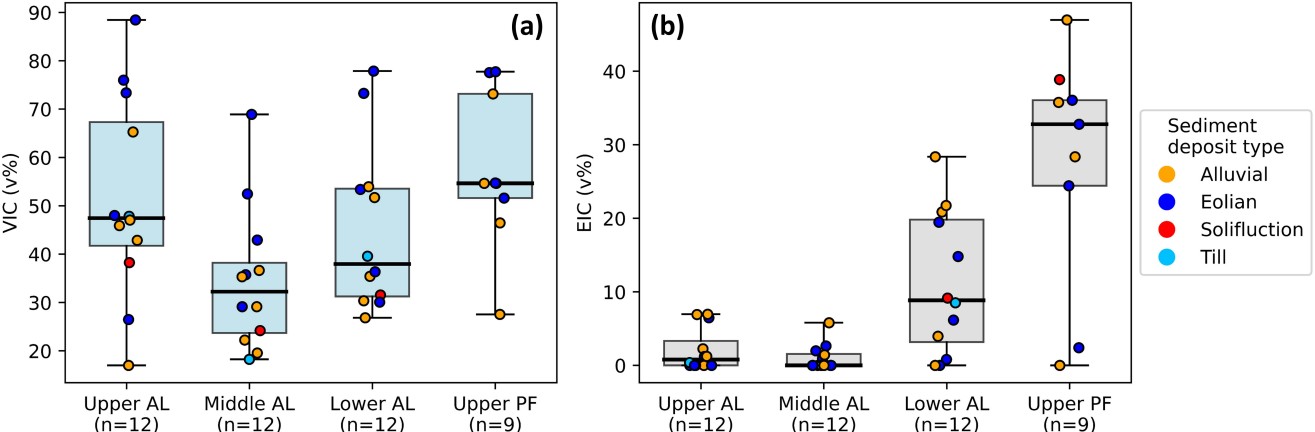

**Figure 3:** Boxplots of (a) the volumetric ice content (VIC), and (b) the excess ice content (EIC) for subsections of the active layer (upper third, central third, lower third) and the uppermost permafrost. AL stands for active layer and PF for permafrost. The number of sites which contributed to each subsection is indicated by n. Three cores did not reach far into the permafrost, and the mean depth of the remaining sites into the permafrost was 0.36 m. Each boxplot illustrates the median (central line), the 25th to 75th percentile range (box), and the full range of observed values (whiskers), with individual data points superimposed. Each point represents one coring site, with the colour displaying the sediment deposit type of this site.

The comparison between the measured in-situ ALT and the expected subsidence from pore ice melt revealed a strong negative correlation ($R^2 = 0.71$, Pearson's r = -0.84, Fig. 4A). Conversely, there was no correlation between ALT and the expected subsidence from only excess ice melt and drainage (Fig. 4B). Overall, there was a poor correlation between the ALT and the total expected subsidence ($R^2 = 0.03$, Fig. 4C). This result suggests that the excess ice (not correlated with ALT, Fig. 4B) has a more significant contribution to the total expected subsidence than the pore ice (correlated with ALT, Fig. 4A). Results from

Section 3.2 confirm this hypothesis. It is important to note that no strong specific patterns associated with the different sediment deposit types were observed (Fig. 4).

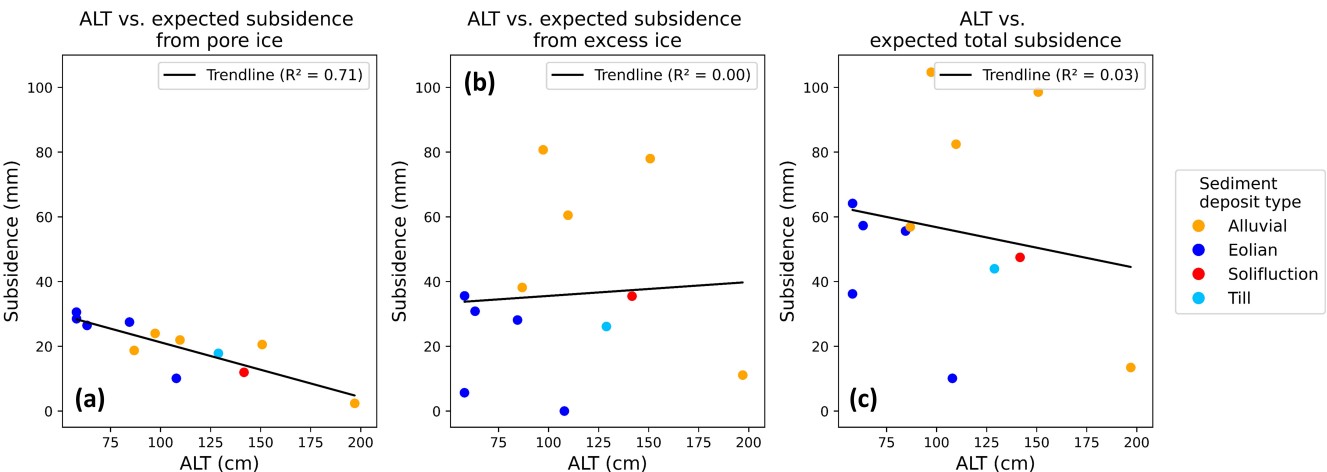

**Figure 4:** Comparison between the active layer thickness (ALT) and the expected subsidence contributions from pore ice melt (a), excess ice melt and meltwater drainage (b) and the combined ice contents (c). Each point represents one site, with the colour displaying the sediment
deposit type of this site.

## 3.2 Comparison of InSAR subsidence and expected subsidence

The seasonal InSAR displacement time series at the coring sites exhibited a wide range in maximum seasonal subsidence, varying from 4 mm to 106 mm, with an average of 57 mm. The InSAR time series began two days after the snow disappeared at the Adventdalen meteorological station, and the InSAR time series displayed seasonal variability throughout the thawing
season 2023 (Fig. 5). The subsidence in the early thawing season was variable across the coring sites, yet generally rather large compared to the middle of the thawing season. In the middle of the thawing season, many sites experienced a decreased subsidence rate. In the late thawing season, the subsidence rate increased again at many, but not all sites. This late-season subsidence aligns with the exceptionally large ALT observed in 2023 (Fig. S8). The onset of the late-season subsidence occurs after a strong precipitation event. At the end of the thawing season in September, the ground surface showed signs of
stabilisation, or exhibited heaving, indicating the beginning of the freeze-up. The observed InSAR subsidence was generally larger at sites with a higher expected subsidence from the in-situ ground ice contents (Fig. 5).

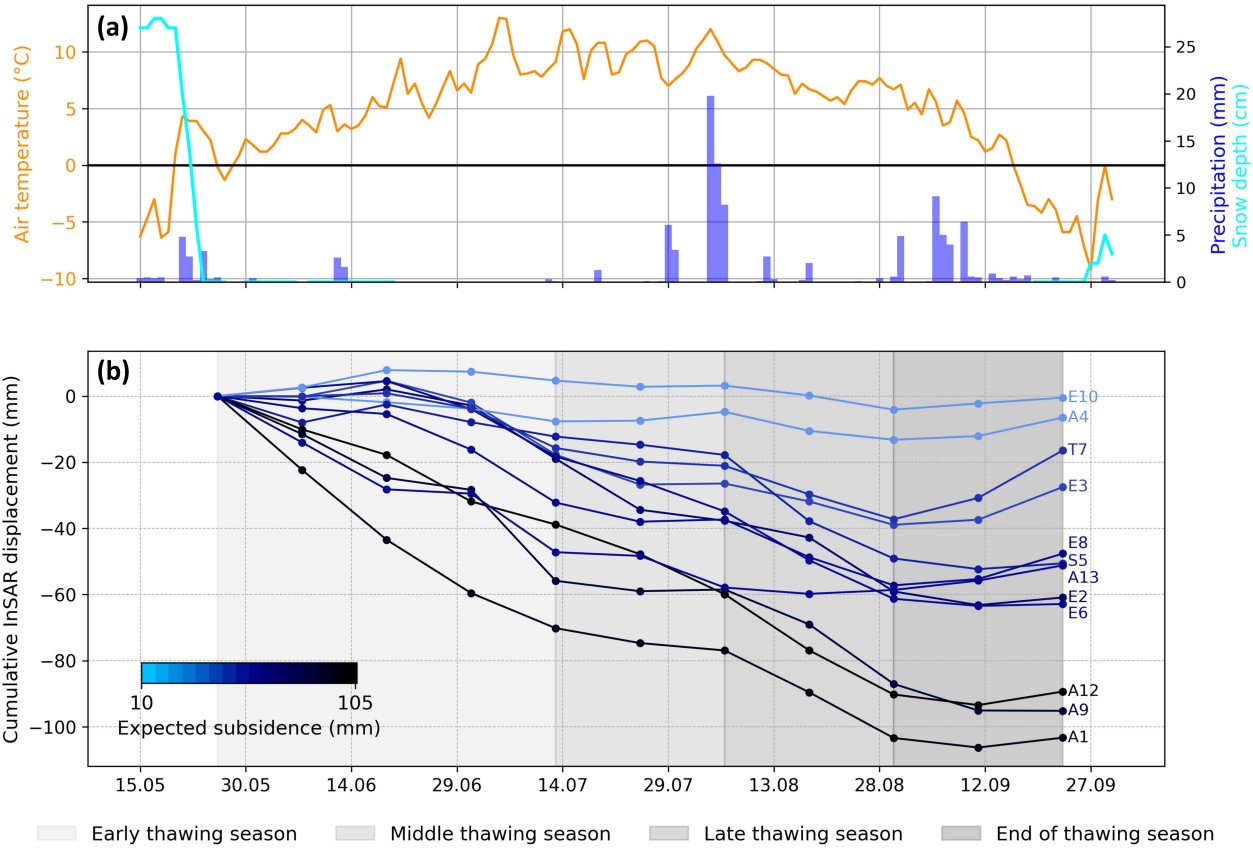

**Figure 5:** Comparison of seasonal InSAR time series 2023 with the meteorological conditions and the magnitude of expected subsidence from in-situ ground ice melt. (a) Meteorological conditions in the thawing season 2023, including snow depth, daily mean air temperature and daily precipitation based on the Adventdalen meteorological station (Norwegian Meteorological Institute, 2024b). (b) Time series of InSAR displacements during the thawing season 2023 from the different coring sites. The line colour displays the expected subsidence at the respective site based on the in-situ ground ice contents and active layer thickness (see Section 2.4). The site name is shown at the end of each time series line, and all sites are described in Table 1.

When assessing the contributions to the expected subsidence from ice melt within the measured ALT, most sites showed a large contribution from excess ice to the expected subsidence signal (Fig. 6). On average across all coring sites, pore ice melt contributed 20 mm to the expected subsidence with a standard deviation (SD) of 8 mm. Excess ice melt contributed an average of 3 mm (SD = 2 mm), and the drainage of meltwater from excess ice contributed an average of 33 mm (SD = 24 mm). The observed maximum InSAR subsidence from summer 2023 aligns well with the expected subsidence when considering both pore ice melt and excess ice melt, including drainage (Fig. 6: grey diamonds).

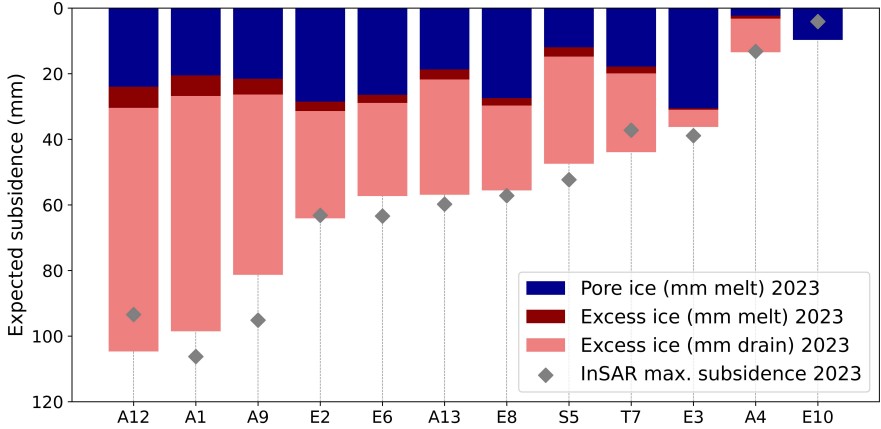

**Figure 6:** Contribution of pore ice melt, excess ice melt, and excess ice meltwater drainage to the total expected subsidence for each coring site. The maximum InSAR subsidence in 2023 is displayed as grey diamonds. The x-label denotes the site names (sediment deposit type and unique core number, see also Table 1).

The comparison between the expected maximum subsidence and the InSAR maximum subsidence shows good overall
alignment (Fig. 7A). The uncertainty in the expected subsidence for the different sites is variable and mostly depends on the excess ice content. A Monte-Carlo simulation incorporating uncertainty over 1000 random iterations resulted in a mean correlation coefficient (r) of 0.82, a mean $R^2$ of 0.68 and a mean absolute error of 15 mm, indicating statistical significance (p-value < 0.01).

However, the correlation between the InSAR subsidence and ALT is weak ($R^2$ = 0.03, r = -0.17, p-value = 0.60) (Fig. 7B).
This aligns with the correlation results between the expected subsidence and the ALT (Fig. 4C).

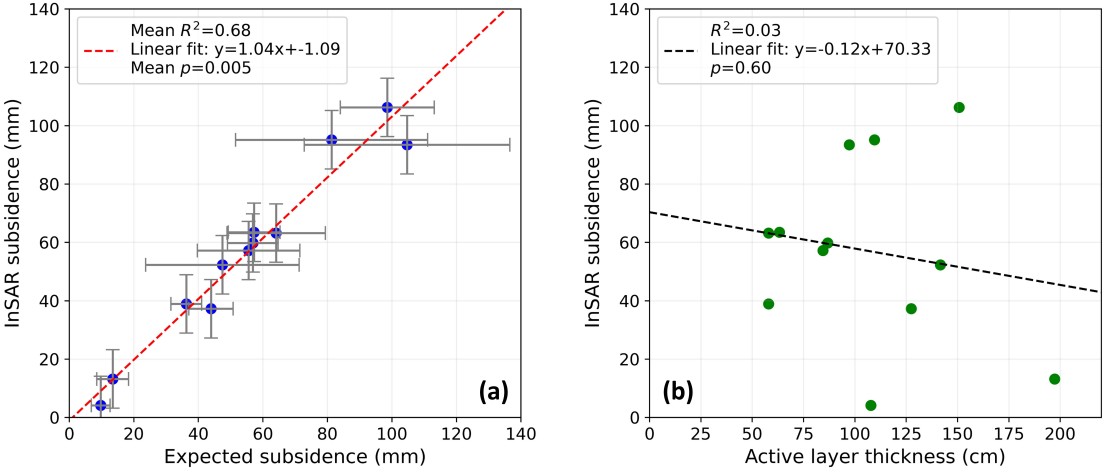

**Figure 7:** Correlation between InSAR subsidence and expected subsidence, as well as active layer thickness (ALT). (a) Correlation between the maximum InSAR subsidence of the thawing season 2023 and the expected subsidence from in-situ ground ice melt (pore ice melt + excess ice melt and drainage). The grey whiskers display the uncertainty in the expected and InSAR subsidence. (b) Correlation
between the maximum InSAR subsidence of the thawing season 2023 and the in-situ ALT.

When evaluating the time series details, the observed InSAR displacements in 2023 align visually well with the ground ice content distribution over depth (Fig. 8). Detailed descriptions of four differing coring sites with varying active layer ice contents follow, providing insights into the spatial diversity observed. Each site's unique characteristics and their impact on both expected and observed subsidence are discussed. All other coring sites are also displayed in a similar format in the Supplement (Fig. S5, S6, S7).

### 3.3 Examples of site-specific geomorphological and subsidence profiles

Alluvial 1 (A1): The A1 coring site is located in an outer inactive alluvial fan (Fig. 1, S1). The active layer is mostly ice-poor, with the dominant sediment type being gravel and the samples falling apart at collection. The lower base of the active layer and the uppermost permafrost are very excess ice rich, due to more fine-grained frost susceptible sediments. According to this ground ice profile, the expected subsidence is very low until the excess ice rich layer is reached. Both in 2021 and 2023, the InSAR subsidence displays a rather strong subsidence from the start of the thawing season, which can only be possible if the thaw front penetrates rapidly through the dry gravel layer of the central active layer. In 2021, the InSAR subsidence levels out earlier than in 2023. In 2023, the InSAR maximum subsidence reaches a similar magnitude as in 2021 and the ground surface stabilizes in July 2023, yet another late-season subsidence pattern starts in August (Fig. 8: Alluvial 1).

Eolian 8 (E8): Located in a depression on a loess terrace within a poorly developed ice wedge polygon (Fig. 1, S1), the E8 coring site features a relatively high VIC partly due to its thick organic layer (0.15 m, Table 1), and the entire core was retrieved intact. Similar to A1, both the lower active layer and the uppermost permafrost at E8 are very excess ice rich. This results in a gradual increase in expected subsidence which then accelerates upon reaching the ice-rich lower layer. The InSAR subsidence in 2021 was less than in 2023. The mean ALT of the adjacent ice-wedge CALM grid was 68 cm in 2021 and 82 cm in 2023. In 2023, the InSAR subsidence shows a slowdown in July, before displaying a further late-season subsidence (Fig. 8: Eolian 8).

Eolian 10 (E10): The coring site E10 is located on a dry, well-drained loess terrace (Fig. 1, S1). The active layer has a low VIC and parts of the sediment core fell apart at collection. Excess ice is minimal in the uppermost permafrost, leading to negligible expected subsidence. Similarly, the InSAR displacement remains around 0 mm throughout the thawing season both in 2021 and 2023 (Fig. 8: Eolian 10). The mean ALT of the surrounding UNISCALM grid was 98 cm in 2021 and 107 cm in 2023, but the ALT difference has no large impact on the subsidence, due to the ice-poor uppermost permafrost.

Alluvial 13 (A13): The coring site A13 is located in an outer sandy alluvial floodplain (Fig. 1, S1). The site has excess ice in the upper active layer, whilst being very dry sand in the lower active layer and uppermost permafrost. As a result of this ground ice profile, the expected cumulative subsidence increases quickly early in the thawing season and then remains constant, which is also observed in the InSAR time series for 2023. In 2021, the thawing onset occurs later, yet again the InSAR subsidence increases quickly before levelling out at a maximum InSAR subsidence similar as 2023 (Fig. 8: Alluvial 13).

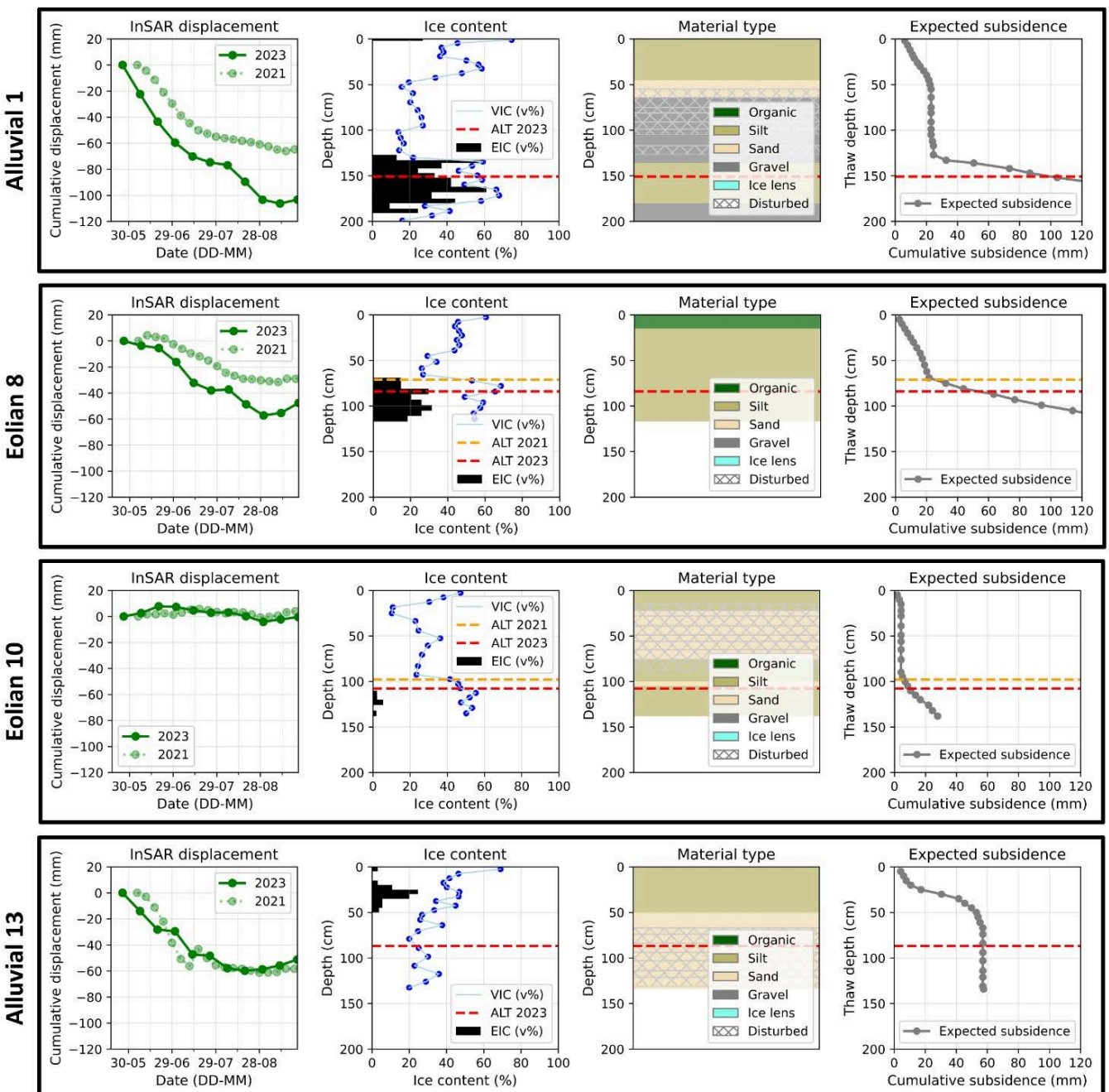

**Figure 8:** Comparison of InSAR time series for selected sites (Section 3.3) for 2023 (warm summer conditions) and 2021 (cold summer conditions) with the different measured types of ground ice and the expected subsidence. Each black rectangle represents one coring site, with the coring site name on the left, and includes four subplots. "InSAR displacement": Cumulative InSAR displacement of the thawing seasons 2021 and 2023. "Ice content": In-situ ground ice contents from spring 2023 displaying the volumetric ice content (VIC) and excess ice content (EIC). "Material type": Material type over depth. "Expected subsidence": The cumulative expected subsidence from in-situ ground ice melt over thaw depth. The red dashed line indicates the active layer thickness (ALT) of the thawing season 2023. The yellow dashed line for site E8 and E10 indicates the ALT of 2021 based on the mean ALT of adjacent CALM grids.

Comparison of 2021 and 2023: As illustrated in Fig. 8, the maximum InSAR subsidence observed in 2021 was consistently lower than in 2023 at all sites featuring excess ice at the transition between the active layer and uppermost permafrost. Conversely, sites with minimal excess ice at the bottom of the active layer and uppermost permafrost exhibited comparable InSAR displacement magnitudes in both years, such as A13. Although no sediment cores were collected in 2021 to confirm the exact ground ice distribution, ground ice in the studied cores from 2023 indicates that excess ice is especially abundant in

the lower active layer and uppermost permafrost (Fig. 3). The 2021 thawing season was exceptionally cold and very shallow ALT were recorded at other stations in Adventdalen, including close to two of our coring sites (Fig. 8, S8). This could indicate that less excess ice melt and drainage contributed to the subsidence under the assumption of similar active layer ground ice conditions. This aligns with the finding that the InSAR subsidence for 2021 matches well the expected subsidence from pore ice melt in 2023, except at alluvial sites with more complex excess ice distributions in the active layer (Fig. S9).

## 4 Discussion

### 4.1 InSAR subsidence as indicator of ground ice melt

Our analysis shows a high correlation between InSAR displacements and in-situ ice contents in the active layer and uppermost permafrost, confirming a good sensitivity of InSAR to ground ice melt. This sensitivity is consistent with results of other studies who have evaluated the use of InSAR for documenting ground water and ice content in permafrost terrain (Chen et al.,

2020; Daout et al., 2020; Wang et al., 2023; Zwieback and Meyer., 2021; Zwieback et al., 2024).

We observed seasonal variations in the InSAR subsidence patterns, which especially align with the distribution of excess ice in the active layer. Pore ice also contributes to the expected subsidence, but in a secondary manner, which more closely aligns with the thaw progression predicted by the Stefan equation (e.g. Fig. 5b: site E3). The expected subsidence contribution from pore ice melt is smaller, since it is only caused by the phase change volume reduction from saturated pores, which were usually

found in the upper and lower active layer. The central active layer was observed to be ice-poor, which is in line with previous field results (e.g. Mackay, 1983; Zwieback et al., 2024). This can be attributed to two-sided freezing of the active layer, with water migrating towards the freezing fronts, which are moving downwards from the surface and upwards from the permafrost table. This caused ice lenses to form especially in the lower active layer (Taber, 1930), and the upper and lower active layer to be saturated with pore ice (Derek and Miller, 1966; Bai et al., 2020), whilst the central active layer remains dry (French,

2007a).

Many sites show a period of low InSAR subsidence in the middle of the thawing season. Yet, the cumulative subsidence at which the InSAR time series stagnates generally does not correspond to the central active layer (inferred from the cumulative expected subsidence). This observation could suggest that the thaw front quickly penetrates the central part of the active layer, and then slows down when it reaches the ice-rich bottom of the active layer. The inverse relationship between active layer

thaw rate and ice content has been previously confirmed (French, 2007a).

The period of stagnating subsidence rate is most pronounced between the 24 July and 08 August 2023 at several of the studied sites, all of which have an excess ice-rich uppermost permafrost. The ground surface stabilization could have been caused by the thaw front reaching the ice-rich transient layer or uppermost permafrost (Shur et al. 2005). Here, sensible and latent heat effects of warming and melting the excess ice, as well as slow drainage of the excess ice meltwater, could have delayed further

surface subsidence, thereby causing a stagnation in the InSAR subsidence pattern. Alternatively, a significant rainfall event (40 mm) during August 4–6, 2023 (Fig. 5), might have temporarily masked subsidence signals due to ground swelling. The rain event could on the other hand also have enhanced the late-season subsidence by percolation of rainwater advecting heat into the lower active layer (Douglas et al., 2020; Magnússon et al., 2022).

The expected subsidence from pore ice melt lies within a plausible range, yet our results indicate that excess ice melt and

meltwater drainage can significantly dominate the expected subsidence signal (Fig. 6: A1, A9, A12). Zwieback and Meyer (2021) mapped excess ice-rich top of permafrost distributions based on the presence or absence of late-season subsidence in an exceptionally warm year in northwestern Alaska within continuous permafrost. They found that the presence of late-season subsidence in InSAR time series closely matches in-situ observations of excess ice-rich permafrost areas. The attribution of late-season subsidence in a warm thawing season to excess ice melt of the uppermost permafrost has also been previously

hypothesized by Bartsch et al. (2019). The findings of Zwieback and Meyer (2021) align with our results, which indicate that excess ice can be the major contribution to the InSAR late-season subsidence signal (Fig. 8, S5-7). Nevertheless, our results also indicate that pore ice contributes to the expected subsidence. Thus, converting the InSAR subsidence to excess ice profiles, such as in Zwieback et al. (2024), will neglect these contributions.

Previous in-situ surface displacement evaluations have been conducted in Siberia by Antonova et al. (2018) and in NW-Canada

by Gruber (2020). Antonova et al. (2018) evaluated InSAR displacements with in-situ surface subsidence from reference rods anchored in permafrost in a Yedoma landscape of the Lena River delta in Siberia. They found that their InSAR displacements from X-band SAR generally underestimated the in-situ subsidence. Compared to this study, our results are based on C-band SAR and could thus be more robust to decorrelation and aliasing (Wang et al., 2020). Gruber (2020) measured in-situ surface displacements with a tilt-arm apparatus in warm permafrost of black forests and peatlands in NW-Canada. They found that the

surface only subsided during the thawing season at a silty site. In two other sites of Gruber (2020), with very thick dry peat layers, the surface heaved, even during the thawing season. Our results cover only one site which has an organic layer throughout the active layer and this site is fully saturated. We do not see a heave signal, but instead a subsidence magnitude that aligns with the pore ice content. Nevertheless, the results from Gruber (2020) indicate that our findings might not apply in areas with different ground conditions, such as thicker vegetation cover and dry peat layers.

Our results also indicate that InSAR subsidence variations within identical sediment deposit types can delineate differences in active layer and top of permafrost ground ice contents. In alluvial fans, the in-situ ground ice content in the upper, drained, coarse-grained areas is very low. In contrast, the outer areas of the fan consist of both coarse and fine-grained layers, with the latter being rich in excess ice. The InSAR time series map these gradients in ground ice content of the active layer and uppermost permafrost, as illustrated for coring sites A1 and A4 (Fig. 1, 8, S5). The presence of excess ice in the active layer

appears to be influenced by several factors, including drainage, landform history, and sediment grain size. In alignment with previous research (e.g. French, 2007b; Cable et al., 2018), excess ice is observed predominantly in fine-grained sediments, whereas it is nearly absent in the gravel and sandy core sections of our retrieved sediment cores (Fig. S10).

Our data exemplify that, even within the same type of sediment deposit, excess ice presence can vary in the active layer and uppermost permafrost. For instance, eolian fine-grained loess terraces show significant variability: some have very low ground ice contents and lack excess ice (e.g., coring site E10), while others have large ground ice contents, particularly in the lower active layer (e.g., coring sites E2, E8) (Fig. 8, S5). This is likely related to drainage and grain size variations (O'Neill et al. 2025), as well as the site-specific formation history of the sediments (Gilbert et al. 2018). The InSAR maps can capture these variations (Fig. 1). Future geomorphological studies could leverage InSAR datasets to enhance ground ice mapping within periglacial landforms by integrating knowledge of sediment grain size and soil moisture conditions. Seasonal variations in InSAR magnitude could be exploited to map active layer ice content (during cold thawing seasons) and top of permafrost ice content (during exceptionally warm thawing seasons). The time series analysis could utilize periods of subsidence stagnation as potential indicator for reaching excess ice rich sediment layers. This approach could help to discriminate different typical landform subsidence patterns, thereby enabling InSAR remote sensing to provide insights into the ground stratigraphy in complex periglacial environments.

## 4.2 Implications for ALT estimation from InSAR subsidence

Previously, InSAR subsidence has been used for ALT inversion by assuming a positive relationship between InSAR subsidence observations of pore ice melt and ALT, with no consideration of excess ice contributions (Liu et al., 2012; Schaefer et al., 2015; Jia et al., 2017; Wang et al., 2018; Peng et al., 2023; Scheer et al., 2023). Some of these studies show good alignment between the InSAR-ALT and ALT from in-situ field measurements (Schaefer et al., 2015). However, recent findings show that the positive correlation between InSAR subsidence observations of pore ice melt and ALT used as basis for the inversion is not valid in drier regions, where the correlation is negative (Chang et al., 2024).

We also find a negative correlation between ALT and expected pore ice subsidence (Fig. 4A), indicating that larger ALT can occur at sites with less pore ice. In addition, our results suggest that traditional inversion models are not universal and might not apply in areas with complex ground stratigraphy. Adventdalen is a periglacial valley which has a very diverse geomorphology with a variety of landforms, some of which have abundant excess ice in the active layer, as for example observed in outer alluvial fans (Fig. 6).

Our results indicate that the surface subsidence signal can be dominated by excess ice melt and drainage instead of pore ice melt, which complicates inversions from thaw subsidence to ALT. Further, the correlation between ALT and the InSAR subsidence was close to random in our results. Considering only pore ice melt to explain the observed subsidence in our study area would cause large errors, since the main subsidence contribution is from excess ice (Fig. 6).

In alignment with our observations (Fig. 7a), Antonova et al. (2018) also observed a poor match between ALT and subsidence. They compared in-situ ALT measurements to in-situ surface subsidence from reference rods anchored in Yedoma permafrost

in Siberia and found a weak positive correlation, with a low coefficient of determination. Further, when comparing the measured in-situ subsidence to the expected subsidence predicted by a pore ice melt model based on soil moisture measurements, they found a moderate alignment but noted significant outliers where the model underestimated the measured subsidence. Very large in-situ subsidence was for example measured in drained lake basins, yet not consistently.

Such outliers in drained lake basins have also been reported in InSAR-ALT estimates. Schaefer et al. (2015) reported outliers in InSAR-ALT estimates over drained lake basins at Barrow, Alaska, where high subsidence led to a large InSAR-ALT under the assumption of only pore ice melt contributing to the subsidence. Similar patterns of high subsidence in such landforms were also reported by Liu et al. (2014) on the Alaskan North Slope near Prudhoe Bay, by Strozzi et al. (2018) at Teshekpuk lake, Alaska, and by Bartsch et al. (2019) in central Yamal, Russia. Our results suggest that the mismatch between InSAR-derived ALT and in-situ ALT is likely due to the omission of excess ice, which has been previously detected in these landforms (Jorgenson and Shur, 2007; Bockheim and Hinkel, 2012; Kanevskiy et al., 2013).

Overall, our study indicates that the contribution from excess ice should not be neglected in models utilizing InSAR time series for active layer characterization, and that simple active layer models relying on constant pore ice contents are oversimplistic in periglacial environments like Adventdalen, Svalbard. Future work should investigate integrating InSAR time series with numerical models that simulate ice segregation processes and excess ice content (e.g., Aga et al., 2023). This could help constrain model parameters or improve process representation by leveraging InSAR-derived surface deformation patterns, potentially through data assimilation techniques (e.g., Aalstad et al., 2018).

## 4.3 Limitations

Our evaluation is limited by the point-wise core measurements versus the much larger InSAR pixel (~18 x 28 m spatial resolution). Due to the inherent variability in periglacial landscapes, this could have introduced artifacts into our analysis (e.g. Hinkel and Nelson, 2003). This effect was moderated by choosing coring locations within a homogenous surrounding, both by looking at the spatial distribution of InSAR displacements and existing maps of sediment deposit types and landforms. However, other studies have displayed the high spatial variability of ground ice contents and surface displacements (e.g., Antonova et al., 2018; Cable et al., 2018; Zwieback et al. 2024). Within our study area, previous studies have displayed variability in surface displacements even within the same sediment deposit type (Rouyet et al. 2019), which is also visible in our data from 2023 (Fig. S11).

The cores were collected at the end of the freezing period, yet water infiltration at the thaw season onset into the frozen part of the sediment column could have caused aggradational ice growth after core collection (e.g. Mackay, 1983, Scherler et al. 2010). This may have caused an underestimation of the in-situ ground ice contents. The excess ice contents are likely conservative, since air bubbles within the excess ice cause an even larger volume loss upon thawing, which is not measured by assessing the supernatant water volume (Morse et al., 2009). Additionally, the excess ice content measurements were performed for 1-7 cm length core sections, which might sometimes have contained ice lenses and then unsaturated sections. These two contributions may have cancelled each other out, thereby reducing the amount of excess ice measured.

Our evaluation is further limited by the small sample size. Both more coring sites and several cores collected at one site would have increased the robustness of our study. Future research should expand the set of available validation data and apply a similar procedure to other permafrost environments to further validate our findings. Complementary ground temperature measurements could also provide more detailed insights into the relations between thaw front propagation, ground ice contents and the InSAR displacement signal.

The InSAR analysis is based on few interferograms in 2023. Due to the failure of Sentinel-1B, the minimum temporal baseline was 12 days, and because of the high subsidence rate, it was only possible to include 24-day interferograms at the end of the thawing season. Nevertheless, the magnitude of the InSAR 2023 dataset is in line with previous years, although a stronger subsidence is observed in 2023, which aligns with the exceptionally warm summer and the very wet conditions (Fig. 1B). In-situ borehole and CALM grid ALT measurements confirm the exceptional large ALT in 2023 (Fig. S8).

The InSAR displacements were converted to vertical displacements by assuming no horizontal movement. Two coring sites (T7 and S5) are placed on 6° and 8° slope angle, which could cause slight horizontal displacement components, for example through solifluction (Harris et al. 2011). Further, the site E8 is located within a low-centre ice-wedge polygon. Whilst the core was extracted at the polygon centre, the InSAR pixel is large enough to include thaw subsidence effects from the ice wedge troughs (Short and Fraser, 2023). Since the ice wedge tops in adjacent polygons have been observed to be located just below the active layer (O'Neill et al., 2025), the late-season subsidence observed at this site might partly reflect this (Burn et al., 2021). Another site, E2, also displays polygonal features indicative of ice wedges. Yet, these sites do not present as outliers in our analysis (Fig. 6).

The InSAR time series starts two days after the snow melt-out date at the Adventdalen meteorological station. Sentinel-2 imagery confirms that all sites except coring site S5 were snow-free or had mixed pixels at the start of the InSAR time series, suggesting minimal impact of snow cover. However, initial subsidence just after snowmelt may not be fully captured at some sites, which could affect comparability. At coring site S5, InSAR subsidence remained negligible until after local snowmelt, which occurred around 7 June 2023.

The reported InSAR uncertainty of 1 cm originates from studies conducted outside the Arctic and may be larger in our study area. However, careful checks for unwrapping errors and decorrelation were conducted as part of the SBAS analysis to mitigate these uncertainties. Interferograms affected by unwrapping errors were manually discarded.

Our expected subsidence calculation is sensitive to the exact ALT, since the bottom of the active layer is most ice rich. In this study manual probing was employed, which allows a more widespread coverage of the pixel, yet is dependent on rather fine-grained ground conditions. Additional borehole temperature measurements could have aided in determining the ALT and provided thaw progression measurements throughout the thawing season, which would have been valuable for comparison against the InSAR subsidence progression.

Lastly, pore ice melt from core sections that were retrieved intact and appeared saturated was considered to contribute to the expected subsidence in this study. However, open-system behavior of the active layer could have allowed lateral or vertical escape and expansion pressure from phase change to dissipate, leading to less pore-ice-melt-induced subsidence than the

theoretical 8% volumetric reduction. This may have caused an overestimation of the expected subsidence from pore ice melt and could have affected the validation assessment between InSAR and expected subsidence. Future research should aim to better quantify the role of pore ice in heave-subsidence soil mechanics under natural freezing and thawing conditions.

## 5    Conclusion

In this study, we compared seasonal SBAS InSAR displacements from C-band SAR with the expected thaw subsidence derived from in-situ ground ice content and thaw depth observations. Sediment cores were retrieved from 12 locations across various periglacial landforms in Adventdalen valley, Svalbard. To estimate the expected subsidence, we calculated pore ice melt, excess ice melt, and excess ice meltwater drainage based on high-resolution measurements of volumetric and excess ice content from the collected sediment cores.

Our findings indicate a good agreement between the expected maximum seasonal thaw subsidence and the maximum seasonal InSAR subsidence. Excess ice melt and drainage dominated the expected subsidence at most sites, which can be attributed to an exceptionally large ALT in the warm summer of the year 2023, promoting thaw into the ice-rich base of the active layer and top permafrost. The difference in InSAR subsidence between the cold thawing season of 2021 and the warm thawing season of 2023 was most pronounced at sites with an ice-rich uppermost permafrost.

We found only a weak correlation between active layer thickness and both expected subsidence and InSAR subsidence for the thawing season of 2023. This may be attributed to the complex distribution of active layer ice contents across our coring sites and the influence of excess ice on the subsidence signal.

Our results have implications for estimating active layer thickness from InSAR. We show that the estimation of active layer thickness from InSAR can be strongly affected by excess ice, although it is often not considered in current retrieval approaches. The choice of a cold thawing season might reduce the likelihood of thawing into the excess-ice-rich transient layer, thus increasing the ability to translate InSAR subsidence to only pore ice melt.

Overall, our results demonstrate that InSAR displacements correspond well with in-situ active layer ground ice contents, supporting the potential future use of InSAR to map seasonal ice content changes and monitor long-term ground ice loss. Ground ice information is often very sparse and extremely important to understand and monitor the consequences of climate change in periglacial environments.

## 545    Appendices

### A.    Uncertainty estimation of expected subsidence

The uncertainty in the expected subsidence $\delta\tau$ was calculated by propagating the individual measurement uncertainties. The uncertainty $\delta V_f$ in the frozen volume of a half cylinder is calculated as:

$$\delta V_f = \sqrt{(0.25\pi d^2 l \cdot \delta d)^2 + (0.125\pi d^2 \cdot \delta l)^2} \qquad \text{[for intact core sections]} \qquad (A1)$$

where $\delta d$ and $\delta l$ are the respective uncertainty in diameter and length, each being $\pm\, 0.2$ cm based on repeated measurements. The total relative error $\delta$ of the VIC, EIC and PIC for each core section can be propagated under the assumption of independence as:

$$\delta VIC_{\text{section}} = 1.09 \cdot \sqrt{\frac{V_f^2 * (\delta M_d^2 + \delta M_w^2) + (M_d - M_w)^2 * \delta V_f^2}{V_f^4}} \tag{A2}$$

$$\delta EIC_{\text{section}} = 1.09 \cdot \sqrt{\frac{V_f^2 * \delta V_{sw}^2 + V_{sw}^2 * \delta V_f^2}{V_f^4}} \tag{A3}$$

$$\delta PIC_{section} = \sqrt{\delta VIC^2 + \delta EIC^2} \tag{A4}$$

Here, the dry weight uncertainty and wet weight uncertainty is $\pm\, 0.01$ g according to the scale accuracy. The supernatant water volume uncertainty $V_{sw}$ is $\pm\, 1$ ml for volumes $\geq 2$ ml. Inserting these uncertainties into the expected subsidence formula for each core section leads to:

$$\delta\tau_{section} = \sqrt{(0.08 * \delta PIC)^2 + (1 * \delta EIC)^2} \tag{A5}$$

which is then summed over the whole ALT:

$$\delta\tau_{measurements} = \sqrt{\sum_0^{ALT} \delta\tau_{section}^2} \tag{A6}$$

In addition to propagating the individual measurement uncertainties, the uncertainty in the in-situ ALT was also considered and estimated as $\pm\, 5$ cm based on the variability in measured ALT at each site ($\delta$ALT). The effect of $\delta$ALT on the expected subsidence was calculated by assessing the mean change in expected subsidence due to a $\pm\, 5$ cm variation in ALT:

$$\delta ALT_{effect} = \left| \frac{\tau_{ALT + \delta ALT} - \tau_{ALT - \delta ALT}}{2} \right| \tag{A7}$$

where $\tau_{ALT \pm \delta ALT}$ represents the recalculated expected subsidence for the adjusted ALT values. This additional uncertainty was then integrated into the total uncertainty of the expected subsidence $\delta\tau$:

$$\delta\tau = \sqrt{\delta\tau_{measurement}^2 + \delta ALT_{effect}^2} \tag{A8}$$

**Data availability**

The ground ice content and active layer thickness data as well as the InSAR time series are archived on Zenodo (https://doi.org/10.5281/zenodo.11187359, Wendt, 2024).

**Author contributions**

LW: Conceptualization, Methodology, Investigation, Formal Analysis, Data Curation, Visualization, Writing - Original Draft, Writing- Reviewing and Editing, Funding acquisition.

LR: Conceptualization, Supervision, Resources, Writing- Reviewing and Editing, Funding acquisition.

HHC: Conceptualization, Supervision, Resources, Writing- Reviewing and Editing, Funding acquisition.

TRL: Data Curation, Writing- Reviewing and Editing, Funding acquisition.

SW: Conceptualization, Supervision, Writing- Reviewing and Editing, Funding acquisition.

**Competing interests**

The authors declare they have no conflict of interest.

**Acknowledgements**

The authors acknowledge funding by the Research Council of Norway (Arctic Field Grant, project no. 342203, RiS ID 12143, 2023), the ESA CCI Permafrost project (4000123681/18/I-NB, 2023) and the Department of Geoscience, University of Oslo. We acknowledge the financial support of the Ministry of Climate and Environment (kap. 1747, post 70) to the Fram Center

PermaRICH project, as well as the funding of the Norwegian Space Agency to the InSAR Svalbard project (Post 74, contract number: 74CO2301), with support from the Geological Survey of Norway.

The research was conducted as part of the Master thesis of Lotte Wendt at the University of Oslo and at the University Centre in Svalbard. We thank Jakob Reif, Leonie Kommerell, Vera van der Veen, Luke Simmons, Mederic Lorry and Knut I. Tveit for assistance with the fieldwork and laboratory analysis. We thank Yngvar Larsen and Heidi Hindberg for their development

on the GSAR processing chain and support to apply it.

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
