# Peer review of "InSAR sensitivity to active layer ground ice content in Adventdalen, Svalbard"

_EGUsphere, 2024_

## Referee Comment (RC1)

This paper compares remotely sensed seasonal ground surface displacement, measured using the InSAR technique with Sentinel-1 (C-band) data, to soil texture and ground ice profiles obtained from 12 core samples in contrasting landform locations in Adventdalen, Svalbard. While there is a rapid increase in studies dealing with permafrost InSAR (particularly SBAS-type), field validation studies remain limited. This research offers significant progress in explaining InSAR spatial variations using detailed frozen-ground core analysis on a watershed scale. I am glad to see the conclusion drawn from your InSAR and in-situ investigations. Your work goes beyond merely addressing the oversimplification of AL-subsidence models; it highlights a critical oversight—the neglect of classic frost heave studies and the role of excess ice in such models. Although I identified several weaknesses and limitations, I support the publication of this paper following necessary revisions outlined below.
* * *
**<Cause of Frost Heave/Seasonal Thaw Settlement and Ice Content Values>**

Classic studies on frost heave mechanisms (e.g., Taber, 1930) indicate that frost heave is primarily caused by water redistribution and ice segregation during soil freezing, while the contribution from water expansion due to phase change is generally negligible in most natural surface ground layers (e.g. Bai et al., 2020). Consequently, seasonal thaw settlement should be explained mainly by the thickness of ice lenses (i.e., excess ice). Discussions on frost heave or thaw settlement based on a model of water phase expansion/shrinkage are therefore inconsistent with these fundamental principles.

While the manuscript acknowledges that pore ice melt is a secondary factor contributing to thaw settlement, its estimated contribution of over 20% to total subsidence (as shown in Fig. 6) may have been overestimated. The formulation in Eq. (4) is not based upon the above-mentioned classical frost heave works and requires further justification. Could you provide references or evidence supporting such a high contribution from pore ice melt?

I suggest revising the Introduction, Discussion, and Conclusion sections to incorporate these considerations and align with established frost heave theories. Additionally, the EIC values in Fig. 8 and S5 require verification. For instance, certain upper horizons of silty soil with high VIC (e.g., A1/E9/E10/A12) show minimal or no excess ice, which seems unusual to me. Similarly, some EIC values in transient layers match VIC levels (e.g., A1/E2/E3/S5/E9/A12), which is strange. If available, could you include cross-sectional photos of the cores to validate these observations?
* * *
As acknowledged in the manuscript, a key limitation of this study is the low representativeness of the in-situ data, with only one core analysis being used to represent a much larger footprint of remote sensing data. Furthermore, the uncertainty discussion focuses solely on InSAR displacement measurements and ice content measurements from the sample cores. The cited 1 cm InSAR uncertainty originates from studies in mid- to low-latitude regions, which differ significantly from Arctic frozen ground conditions. In Arctic environments, issues such as decorrelation and unwrapping errors are exacerbated by surface moisture, snow cover, and

vegetation influences. Additionally, the spatial heterogeneity of ground ice content in Arctic regions is far greater than what might be inferred from similar depositional conditions.

To strengthen the discussion, I suggest incorporating spatial variations of InSAR subsidence (categorized by sedimentation-based land classifications), ALT measurements (CALM data and your observations), and ground ice content (potentially derived from literature). This would provide a more comprehensive perspective on the uncertainties and enhance the paper's robustness.

For the InSAR analysis, it would be beneficial to elaborate on how highly decorrelated interferograms were handled. Please clarify the threshold coherence values and justify the acceptance of a maximum temporal baseline of 24 days. Including a summary of coherence statistics for the accepted interferograms would greatly aid reproducibility and facilitate future comparisons.

This additional information would address current gaps in the uncertainty discussion and provide critical context for interpreting the results.

**<Method>**

L. 125: How many points were measured for ALT? Could you include statistics in Fig. S3?

L. 128: Why is it necessary to adjust the original ground surface height based on a single-year core analysis? Frost heave caused by ice segregation can vary significantly from one winter to the next, depending on climatic and hydrological conditions. For seasonal analyses, it would be more appropriate to base the ground surface level on the moment of maximum thaw, as this approach minimizes the influence of interannual variability in frost heave. However, such an adjustment could be justified if the primary focus is on inter-annual subsidence trends, such as those driven by thermokarst development or long-term permafrost degradation.

**(Section 2.3 Analysis of Ground Ice Content)**

How was the volumetric ice content (VIC) determined for disturbed samples? Please provide a detailed explanation of the method used to measure the volume of sediments and supernatant water. This additional detail would help clarify the accuracy and reliability of the VIC measurements.

I find it difficult to agree with the interpretations in L. 346–347 and L. 371–373, as they appear to rely on the assumption that water phase-change expansion contributes significantly to ground ice dynamics. This assumption is inconsistent with established studies, which typically suggest that water phase expansion plays a negligible role compared to ice segregation processes. Could you provide further clarification or supporting evidence for this claim?

Additionally, the approximate 8% contribution from Liu et al. (2012) is mentioned but not sufficiently explained. Please elaborate on how this value was derived and its specific relevance to your study.

**<Additional Suggestions and Corrections>**

L. 256: How exceptional were the 2023 ALT values? Could you compare them with previous years' data, if available?

L. 269–270: Please confirm the accuracy of these numbers, as they seem inconsistent with related data.

L. 357: Clarify whether "stagnating subsidence" refers to stabilization of the ground surface when the thaw front passes through ice-poor layers or ice-rich layers. Is your "excess ice-rich base" top or bottom of the ice-rich layer at the ALT-permafrost boundary?

L. 360: Frost heave during mid-summer heavy rain under sustained high air temperatures (>5°C for 1–2 months) seems unrealistic. Could alternative explanations, such as vegetation growth or instrument malfunctions, account for the reported thaw-season heave (1–2 cm)?

L. 389–391: If the primary focus of your paper is excess ground ice and its relationship to seasonal thaw settlement or frost heave, I strongly recommend revisiting foundational studies on frost heave mechanisms and long-term development of ice-rich layer in the boundary zone between AL and permafrost, such as those by Taber (1930) and Shur et al. (2005).

L. 395: One of the main factors to determine the amount of segregation ice is soil moisture. Lateral drainage is one component that controls soil moisture through water balance in the active layer.

L. 398: Replace "grain size" with "grain size distribution of mineral soil particles." Also, consider using "soil moisture conditions" instead of "drainage conditions," as good drainage place can still maintain high soil moisture levels depending on local hydrological conditions.

L. 400–401: This statement could be reconsidered to align with established observations. Thaw settlement typically stagnates when the thaw front passes through zones with minimal ice lenses (segregation ice) and intensifies when progressing through ice-rich layers.

L. 407-409: This sentence could benefit from clarification. Could you elaborate on what is meant by "increase the influence on"? If you are referring to processes at the interface between the active layer and uppermost permafrost, I suggest adopting the concept of the "Transient Layer" as described by Shur et al. (2005).

L. 412–413: "Negative" should be replaced with "positive"? It seems to be a contradiction with the subsequent sentence. Please revise for consistency.

L. 416–418, 425–428: The ALT inversion described in these references is based on an incorrect frost heave theory that attributes ground movement to water volume expansion upon freezing rather than to water redistribution caused by ice segregation. Consider revising these interpretations to align with established frost heave mechanisms.

L. 415: The alignment between in-situ ALT probing and temperature measurements is not entirely clear. Could you elaborate on what this alignment reveals and how it connects to the following part of the sentence?

L. 428: Correct "Teshepuk" to "Teshekpuk."

L. 432: I am very pleased with the conclusion drawn from your InSAR and in-situ investigations. Your work goes beyond merely addressing the oversimplification of AL-subsidence models; it highlights a

critical oversight—the neglect of classic frost heave studies and the role of excess ice in such models. This omission in the previous works was surprising for me.

**References**

- Taber, S. (1930). *The mechanics of frost heaving.* J. Geol., 38, 303–317.
- Bai, R. et al. (2020). *Investigation on frost heave of saturated–unsaturated soils.* Acta Geotechnica, 15(11), 3295–3306. https://doi.org/10.1007/s11440-020-00952-6
- Shur, Y., Hinkel, K. M., & Nelson, F. E. (2005). *The Transient Layer: Implications for Geocryology and Climate-Change Science.* Permafrost and Periglacial Processes, 16, 5–17.

---

## Author Comment (AC1)

**Answers to Reviewer 1**

**In the following, we answer (shown in blue) point by point the comments from the reviewers (shown in black). Major comments have been numbered in order to refer within our replies to previous answers. Changes to the manuscript are shown in italics and the line numbers refer to the original preprint.**

This paper compares remotely sensed seasonal ground surface displacement, measured using the InSAR technique with Sentinel-1 (C-band) data, to soil texture and ground ice profiles obtained from 12 core samples in contrasting landform locations in Adventdalen, Svalbard. While there is a rapid increase in studies dealing with permafrost InSAR (particularly SBAS-type), field validation studies remain limited. This research offers significant progress in explaining InSAR spatial variations using detailed frozen-ground core analysis on a watershed scale. I am glad to see the conclusion drawn from your InSAR and in-situ investigations. Your work goes beyond merely addressing the oversimplification of AL-subsidence models; it highlights a critical oversight—the neglect of classic frost heave studies and the role of excess ice in such models. Although I identified several weaknesses and limitations, I support the publication of this paper following necessary revisions outlined below.

We would like to thank the reviewer for the detailed comments! These have been very helpful and are addressed point by point below.

Based on feedback from both reviewers, we have added additional supplementary figures, which are currently numbered sequentially after the previously existing ones. In the revised manuscript and supplement, we will update the numbering and order of the supplementary figures to maintain a logical sequence when referring to them in the revised manuscript.

**<Cause of Frost Heave/Seasonal Thaw Settlement and Ice Content Values>**

1.  Classic studies on frost heave mechanisms (e.g., Taber, 1930) indicate that frost heave is primarily caused by water redistribution and ice segregation during soil freezing, while the contribution from water expansion due to phase change is generally negligible in most natural surface ground layers (e.g. Bai et al., 2020). Consequently, seasonal thaw settlement should be explained mainly by the thickness of ice lenses (i.e., excess ice). Discussions on frost heave or thaw settlement based on a model of water phase expansion/shrinkage are therefore inconsistent with these fundamental principles.

    In our study, we are looking at the subsidence signal based on in-situ ground ice contents using accepted/standard methodology according to Everdingen (1998) to distinguish between pore ice and excess ice/segregation ice. Pore ice is thereby defined as ice occurring in the pores of soils and rocks, which does upon melting not yield water in excess of the pore volume of the same

soil/rock when unfrozen. Excess ice is thereby defined as segregated ice in excess of the pore space, which consists of ice lenses from migration of pore water to the frozen fringe.

The distribution of ground ice in our in-situ measurements is the result of in-situ freezing of water in soil pores and of the above-mentioned water migration and ice segregation both forming ice lenses (e.g. Taber, 1930), but also causing saturation of some core sections with pore ice only (usually adjacent to excess ice rich core sections) (e.g., Dirksen and Miller, 1966; Bai et al., 2020). In our laboratory measurements, we assessed whether excess ice was present in small core sections by measuring supernatant water volume upon thawing (L. 150). If so, then this excess ice volume was assumed to cause a 100% thaw settlement due (1) the phase change volume loss (water is ~8 % denser than ice) and (2) drainage of the resulting meltwater. For intact core sections which did not have any excess ice based on our measurements, but which visually appeared saturated with pore ice, we considered phase change volume loss and thus a small settlement was expected (8 % of pore ice volume). This phase change causing small soil volume changes for samples saturated with pore water/ice upon freezing/thawing is also discussed in Bai et al. (2020) and in other studies (Dirksen and Miller, 1966; Derk and Unold, 2023; Dumais and Konrad, 2024). For core sections which were unsaturated with little pore ice (dry and retrieved disturbed), no subsidence signal was expected from volume loss induced by pore ice melt, as we expect the soil matrix to be already fully condensed.

See answer to comment 4 to see the changes performed in the manuscript.

2. While the manuscript acknowledges that pore ice melt is a secondary factor contributing to thaw settlement, its estimated contribution of over 20% to total subsidence (as shown in Fig. 6) may have been overestimated.

Figure 6 does not include any percentage units but shows the subsidence contributions in mm. The values are based on the in-situ/lab measurements and equation 4, as explained in the answer to question 1. We have doublechecked the calculations. The values are, according to our measurements, correct.

The estimated contribution from pore ice melt is based on the phase change volume reduction of pore ice to pore water (8 % volume loss of pore ice volume) in intact, saturated core sections (Dumais and Konrad, 2024).

Your comment highlights the need for method clarifications in the paper. See answer to comment 4 to see the changes performed in the manuscript.

3. The formulation in Eq. (4) is not based upon the above-mentioned classical frost heave works and requires further justification. Could you provide references or evidence supporting such a high contribution from pore ice melt?

The equation 4 builds up on the frost heave observations from Taber (1930). It acknowledges that the measured volume of excess ice will likely cause a 100 % volume loss upon thawing due to (1) volume loss upon melting which is approximately 8 % and (2) subsequent drainage accounting for the remaining 92 % of volume loss.

The additional contribution from pore ice melt in saturated core sections is to account for the 8 % volume loss upon phase transition of the pore ice (Dumais and Konrad, 2024).

See answer to comment 4 to see the changes performed in the manuscript.

4. I suggest revising the Introduction, Discussion, and Conclusion sections to incorporate these considerations and align with established frost heave theories.

We have added the following section to the Introduction in L. 48-53:

> "*Ground ice, which varies in distribution, includes both pore ice within soil pores and excess ice that exceeds the soil's pore space (Everdingen, 1998). Pore ice forms when soil moisture freezes within the existing pore spaces of mineral and organic soils. Depending on the degree of saturation, the phase change causes either a volume expansion within the pore space or, if the pores are saturated with water, an expansion of the soil structure itself due to the pressure exerted by the growing ice. Ice segregation processes can cause the migration of water towards the freezing front, enriching soil pores further with ice and leading to the growth of ice lenses (Derek and Miller, 1966). If the accumulation of ice exceeds the pore space, excess ice occurs (Taber, 1930; Rempel et al., 2007).*
> *The melting of pore ice in saturated ground can lead to thaw consolidation, caused by the volume loss associated with the density difference between ice and water (approx. 8%) (Dumais and Konrad, 2024). The melting of excess ice has an even more pronounced effect, as the loss of ice that exceeds the soil's pore space can cause significant subsidence when the resulting meltwater drains away* (Morgenstern and Nixon, 1971).
> Comprehensive field validation to measure the impact these ice types have on InSAR measurements is still lacking, underscoring the need for improved understanding of how InSAR captures these seasonal ground ice changes (Bartsch et al., 2023)."

We have rewritten section 2.4 in the Methods to better explain and justify the expected subsidence calculation based on previous studies (L. 155-167):

> "*The core sections were retrieved either intact or disturbed. Samples that fell apart during sampling and/or analysis (disturbed) were observed to be very dry and therefore classified as unsaturated and not considered to contribute to the subsidence signal. The intact core sections were observed to be visually saturated with pore ice and contained sometimes also excess ice based on the laboratory measurements. Based on these intact core sections, the expected subsidence was derived.*
> *In the intact core sections, both excess ice and pore ice melt are expected to cause a volume reduction according to the density difference between ice and water (approximately 8 %) (Dumais and Konrad 2024). However, excess ice, as it exceeds the pore space of the soil column, can in addition drain or redistribute into unsaturated pore spaces, likely leading to a full volume loss of the excess ice volume (Morgenstern and Nixon, 1971).*
> Based on this rationale, the expected subsidence $\tau$ for the thawing season 2023 was calculated *from intact core sections*, *consisting of* pore ice melt (8 % volume reduction), excess ice melt (8 % volume reduction) and excess ice meltwater drainage (92 % volume reduction) within the ALT measured at the end of the thawing season 2023:

$$\tau = \sum_0^{ALT} (\underbrace{0.08 * PIC}_{\substack{\text{melt of} \\ \text{pore ice}}} + \underbrace{0.08 * EIC}_{\substack{\text{melt of} \\ \text{excess ice}}} + \underbrace{0.92 * EIC}_{\substack{\text{drainage of} \\ \text{excess ice meltwater}}}) * l \qquad (4)$$

Additionally, *also* the *individual* contributions from the melt of pore ice, excess ice, and the drainage of excess ice meltwater to the total subsidence were calculated. The uncertainty in the expected subsidence was calculated by propagating the measurement uncertainties and is described in Appendix A."

In the Discussion, we have:
- Included a short discussion of likely ice segregation water migration in the active layer considering the observed ice-poor central active layer (L. 347-350):
  "The central active layer was observed to be ice-poor, which is in line with previous field results (e.g. Mackay, 1983; Zwieback et al., 2024). This can be attributed to two-sided freezing of the *active layer, with water migrating towards the freezing fronts, which are moving downwards from the surface and upwards from the permafrost table. This caused ice lenses to form especially in the lower active layer (Taber, 1930), and the upper and lower active layer to be saturated with pore ice (Derek and Miller, 1966; Bai et al., 2020), whilst the central active layer remains dry (French, 2007a).*"
- Extended our discussion of limitations (L. 446-448):
  "The excess ice contents are likely conservative, since air bubbles within the excess ice cause an even larger volume loss upon thawing but are not measured by assessing the supernatant water volume (Morse et al., 2009). *Additionally, the excess ice content measurements were performed for 1-7 cm length core sections, which might sometimes have contained ice lenses and then unsaturated sections. These two contributions may have cancelled each other out, thereby reducing the amount of excess ice measured.*"

5. Additionally, the EIC values in Fig. 8 and S5 require verification. For instance, certain upper horizons of silty soil with high VIC (e.g., A1/E9/E10/A12) show minimal or no excess ice, which seems unusual to me.
   Based on the applied measurement techniques, we confirm that the minimal amount of excess ice is correct in these cases.
   The low excess ice content in the upper soil horizons at these coring sites may be due to rapid freezing. If the freezing front moved quickly downward from the soil surface, water from below may not have had enough time to migrate into the frozen fringe and form ice lenses (Fu et al., 2022). Such differences in freeze front propagation speed might relate to different snow cover between the sites during early winter, which has been observed in-situ.
   We have taken cross-sectional photos of all cores (see supplement Fig. S10).

6. Similarly, some EIC values in transient layers match VIC levels (e.g., A1/E2/E3/S5/E9/A12), which is strange.
   The EIC matches VIC in some cases, as the volume of the core is filled by an ice lens in that section. In those cases, nearly all of the VIC is excess ice.

7. If available, could you include cross-sectional photos of the cores to validate these observations?
   Done. See Fig. S10.

[Figure]

*Figure S10: Cross-sectional photos of all cores. The coring site name is shown on top, and depths (in cm) are increasing from top to bottom.*
* * *
8. As acknowledged in the manuscript, a key limitation of this study is the low representativeness of the in-situ data, with only one core analysis being used to represent a much larger footprint of remote sensing data. Furthermore, the uncertainty discussion focuses solely on InSAR displacement measurements and ice content measurements from the sample cores. The cited 1 cm InSAR uncertainty originates from studies in mid- to low-latitude regions, which differ significantly from Arctic frozen ground conditions. In Arctic environments, issues such as decorrelation and unwrapping errors are exacerbated by surface moisture, snow cover, and vegetation influences. Additionally, the spatial heterogeneity of ground ice content in Arctic regions is far greater than what might be inferred from similar depositional conditions.
To strengthen the discussion, I suggest incorporating spatial variations of InSAR subsidence (categorized by sedimentation-based land classifications), ALT measurements (CALM data and your observations), and ground ice content (potentially derived from literature). This would provide a more comprehensive perspective on the uncertainties and enhance the paper's robustness.

We fully acknowledge the key limitation of the low amount of in-situ data, and have extended our discussion of limitations based on this comment. Unfortunately, no in-situ surface displacement data is available in this area. Also, no other active layer ground ice estimates exist for the study area, so we are not able to further quantify ground ice content uncertainties. We have included the standard deviation of the ALT measurements per coring site in Table 1 to indicate the observed variability per pixel.

We made the following additions to the discussion of limitations:

- L. 438-443: "Our evaluation is limited by the point-wise core measurements versus the much larger InSAR pixel (~18 x 28 m spatial resolution). Due to the inherent variability in periglacial landscapes, this could have introduced artifacts into our analysis (e.g. Hinkel and Nelson, 2003). This effect was moderated by choosing coring locations within a homogenous surrounding, both by looking at the spatial distribution of InSAR displacements and existing maps of sediment deposit types and landforms. However, other studies have displayed the high spatial variability of ground ice contents and surface displacements (e.g., Antonova et al., 2018; Cable et al., 2018; Zwieback et al. 2024). *Within our study area, previous studies have displayed variability in InSAR surface displacements even within the same sediment deposit type (Rouyet et al. 2019), which is also visible in our InSAR dataset from 2023 (Fig. S9).*"

[Figure]

*Figure S9: Boxplot of the maximum InSAR subsidence in 2023 per sediment deposit type in the study area. Each class consists of 2000 randomly chosen InSAR pixels which fall within the sediment deposition type (Rouyet et al., 2019; modified from Härtel and Christiansen, 2014).*

- L. 462: *"The reported InSAR uncertainty of 1 cm originates from studies conducted outside the Arctic and may be larger in our study area. However, careful checks for unwrapping errors and decorrelation were conducted as part of the SBAS analysis to mitigate these uncertainties. Interferograms affected by unwrapping errors were manually discarded."*

9. For the InSAR analysis, it would be beneficial to elaborate on how highly decorrelated interferograms were handled. Please clarify the threshold coherence values and justify the acceptance of a maximum temporal baseline of 24 days. Including a summary of coherence statistics for the accepted interferograms would greatly aid reproducibility and facilitate future comparisons.

We have created coherence time series from each coring site and have added them to the supplement as Fig. S11. We have also updated the InSAR analysis section in L. 176-180:

"To mitigate temporal decorrelation and phase ambiguities from strong subsidence in the exceptionally warm summer 2023, a maximum temporal baseline of 24 days was used. This threshold was chosen *after inspection of interferograms created with longer temporal baselines, which were strongly decorrelated. All interferograms with 12- and 24-day temporal baselines were manually inspected and highly decorrelated interferograms were discarded*. Both ascending and descending geometries cover the study area, but the ascending stack was incomplete due to a missing image in August 2023. After comparing time series constructed from both geometries, the descending time series was selected for further analysis due to *unwrapping errors* in the ascending data set. *The coherence time series of all coring sites based on the further used descending dataset are shown in Figure S11.*".

[Figure]

*Figure S11: (a) Coherence time series for each coring site during the thawing season of 2023. For each date, the coherence is computed at the pixel level by averaging the coherence of all interferograms that include the date. This results in a time series of average coherence per date for each pixel. (b) The InSAR cumulative displacement over the thawing season 2023, with the same colouring per coring site and aligned x-axis as in (a). The mean coherence per coring site is included in the legend. The map in (c) displays the spatial distribution of the mean coherence of the thawing season 2023.*

This additional information would address current gaps in the uncertainty discussion and provide critical context for interpreting the results.

Thank you for these suggestions!

**<Method>**

L. 125: How many points were measured for ALT? Could you include statistics in Fig. S3?

Per coring site, ALT was measured at 9 points spread across the pixel (one measurement at the coring location, and then per cardinal direction one measurement at 10 m distance from the coring location and one at 20 m distance). The mean ALT from these measurements per coring site was used for further analysis. This is explained at L. 126-127. We have now also included the standard deviation of the ALT per site in Table 1.

Additionally, Fig. S3a shows the ALT at 5 other locations in the study area (3 boreholes and 2 CALM sites) for the study years 2023 (warm thawing season) and 2021 (cold thawing season) to give insights into the temporal development of the ALT in the study area. We have updated the legend in this figure to include the number of measurement points of the CALM grids. The updated Fig. S3 is shown on page 11 of this document.

L. 128: Why is it necessary to adjust the original ground surface height based on a single-year core analysis? Frost heave caused by ice segregation can vary significantly from one winter to the next, depending on climatic and hydrological conditions. For seasonal analyses, it would be more appropriate to base the ground surface level on the moment of maximum thaw, as this approach minimizes the influence of interannual variability in frost heave. However, such an adjustment could be justified if the primary focus is on inter-annual subsidence trends, such as those driven by thermokarst development or long-term permafrost degradation.

The ALT measurements were done at the end of the thawing season 2023. The cores were extracted in spring 2023, short before the beginning of the thawing season. If we use the in-situ ALT to know how deep the cores thawed, then we will have an error in the depth of thaw, since the core length is measured in frozen state with ice in the active layer, whilst the ALT is measured in thawed ground, which is more consolidated.

We have updated L. 128-130 to better explain this:

> *"The ALT was measured at the end of the thawing season in 2023, while the cores were extracted before the thawing season when the active layer was still frozen. Since ground ice melt leads to surface subsidence over the thawing season, the ALT measurement did not directly correspond to the core length (O'Neill et al., 2023). To account for this, we applied a correction based on the expected subsidence (Table 1), derived from the core data (see Section 2.4)."*

(Section 2.3 Analysis of Ground Ice Content)
How was the volumetric ice content (VIC) determined for disturbed samples?

For disturbed core sections, the VIC was determined based on the same formula as for intact core sections. This includes the volume of the core section and the weight difference between dry and wet weight to determine the ice volume (L. 147). We have added that the diameter and length of these disturbed sections were based on the core barrel diameter and the length of retrieved sections measured in the field:

> *"For disturbed core sections, the length l was measured in the field during retrieval from the borehole and the diameter d was based on the core barrel diameter."*

Please note that the VIC of disturbed samples was never considered to contribute with a subsidence signal, since the disturbed sections were observed to be unsaturated and thus no expansion from pore ice past the initial porosity is expected (explained in L. 163-165).

Please provide a detailed explanation of the method used to measure the volume of sediments and supernatant water. This additional detail would help clarify the accuracy and reliability of the VIC measurements.

We have now included in line 140 that the diameter and length measurements were done with a ruler. The supernatant water volume was measured based on thawing the sample in a graded measurement beaker and reading of the volume of supernatant water above the settled sediments with additional help of a ruler (added in line 143).

The uncertainties of these measurements are described in Appendix A of the preprint (see L. 492, L. 498).

I find it difficult to agree with the interpretations in L. 346–347 and L. 371–373, as they appear to rely on the assumption that water phase-change expansion contributes significantly to ground ice dynamics. This assumption is inconsistent with established studies, which typically suggest that water phase expansion plays a negligible role compared to ice segregation processes. Could you provide further clarification or supporting evidence for this claim?

Please see our replies to comment 1 and 3 regarding why we consider phase-change volume reductions from pore ice melt to occur for saturated core sections. We have therefore not changed L. 371-373.

Based on both your comment and the feedback from reviewer 2, we have updated L. 346-347 to:

> "Pore ice also contributes to the *expected* subsidence, but in a secondary *manner, which more closely aligns with the thaw progression predicted by the Stefan equation (e.g. Fig. 5b: site E3). The expected subsidence contribution is smaller, since it is only caused by the phase change volume reduction from saturated pores, which were usually found in the upper and lower active layer."*

Additionally, the approximate 8% contribution from Liu et al. (2012) is mentioned but not sufficiently explained. Please elaborate on how this value was derived and its specific relevance to your study.

The 8 % volume reduction of pore ice is based on the density difference between ice (0.917 g/cm$^3$) and water (1 g/cm$^3$). Upon melting of ice, the resulting water will take up an ~8 % lower volume, e.g. $(1-0.917) \times 100 = 8.3$ %. This density difference causing the ~8 % volume loss is explained in the updated section 2.4, e.g.:

> "In the intact core sections, both excess ice and pore ice melt are expected to cause a volume reduction according to the density difference between ice and water (approximately 8%) (Dumais and Konrad, 2024)."*

See also our answer to above comment 4.

**<Additional Suggestions and Corrections>**

L. 256: How exceptional were the 2023 ALT values? Could you compare them with previous years' data, if available?

We do not have ALT measurements from the coring sites for previous years. However, we have added a plot (Fig. S3c) which shows the ALT development at three boreholes and two CALM grids in the study area over a longer period. 2023 was in that period the year with the largest ALT across these sites.

[Figure]

*Figure S3: Comparison of ALT of 2021 (cold thawing season) and 2023 (warm thawing season) from three boreholes and two CALM grids in Adventdalen (a). The location of the boreholes and CALM grids is shown in (b). The long-term development of ALT measurements at these sites and the mean annual air temperature (MAAT) development at Svalbard airport is shown in (c), with 2023 underlain in red. The colours of the sites align between the subplots. The background in (b) is from TopoSvalbard (Norwegian Polar Institute, 2014a).*

L. 269–270: Please confirm the accuracy of these numbers, as they seem inconsistent with related data.

These numbers have been verified, and they align with the expected subsidence contributions in Figure 6 (average across all sites). Please note that the unit of the expected subsidence is mm and not %. Please also note that the melt of excess ice and the subsequent drainage of the resulting meltwater are listed separately (L. 269-270). We have updated L. 268 to say "on average across all coring sites" instead of "on average":

> "*On average across all coring sites*, pore ice melt contributed 20 mm to the expected subsidence with a standard deviation (SD) of 8 mm."

L. 357: Clarify whether "stagnating subsidence" refers to stabilization of the ground surface when the thaw front passes through ice-poor layers or ice-rich layers. Is your "excess ice-rich base" top or bottom of the ice-rich layer at the ALT-permafrost boundary?

We have updated this sentence to clarify that "excess ice-rich base of the active layer" refers to the bottom of the active layer, at the transition to the transient layer, which is very ice-rich at most sites. The updated paragraph is:

> "The period of stagnating subsidence rate is most pronounced between the 24 July and 08 August 2023 at several of the studied sites, all of which have an excess ice-rich uppermost permafrost. *The ground surface stabilization could have been caused by the thaw front reaching the ice-rich transient layer or uppermost permafrost (Shur et al., 2005).* Here, sensible and latent heat effects of warming and melting the excess ice, as well as slow drainage of the excess ice meltwater, could have delayed further surface subsidence, thereby causing a stagnation in the InSAR subsidence pattern."

L. 360: Frost heave during mid-summer heavy rain under sustained high air temperatures (>5°C for 1–2 months) seems unrealistic. Could alternative explanations, such as vegetation growth or instrument malfunctions, account for the reported thaw-season heave (1–2 cm)?

Please note that we did not observe any mid-summer heave in the InSAR time series, only a stagnation in the cumulative subsidence. We questioned if this stagnation might have been caused by other heave effects masking the thaw subsidence.

We agree to remove the hypothesis that ice formation at the freezing front in the lower active layer could have caused a heave signal. However, we still believe that ground swelling could have occurred given the significant rain event. On the other hand, the effect of vegetation growth seems unrealistic, since all sites have very low growing vegetation (mosses or barren). We have updated this section to:

> "Alternatively, a significant rainfall event (40 mm) during August 4–6, 2023 (Fig. 5), might have temporarily masked subsidence signals due to ground swelling. The rain event could on the other hand also have enhanced the late-season subsidence by percolation of rainwater advecting heat into the lower active layer (Douglas et al., 2020; Magnússon et al., 2022)."

L. 389–391: If the primary focus of your paper is excess ground ice and its relationship to seasonal thaw settlement or frost heave, I strongly recommend revisiting foundational studies on frost heave mechanisms and long-term development of ice-rich layer in the boundary zone between AL and permafrost, such as those by Taber (1930) and Shur et al. (2005).

We have incorporated these studies at other locations in the paper (see also answers to comment 4). We decided to keep this sentence as it is, as it serves as introduction statement to the following paragraphs to discuss how drainage, landform history and sediment grain size influence the excess ice presence in the active layer.

L. 395: One of the main factors to determine the amount of segregation ice is soil moisture. Lateral drainage is one component that controls soil moisture through water balance in the active layer.

Agree. We have updated this sentence by replacing "drainage variations" with "soil moisture variations".

L. 398: Replace "grain size" with "grain size distribution of mineral soil particles." Also, consider using "soil moisture conditions" instead of "drainage conditions," as good drainage place can still maintain high soil moisture levels depending on local hydrological conditions.

We have replaced "grain size" with "sediment grain size" and "drainage conditions" with "soil moisture conditions". The resulting sentence is now:

*"Future geomorphological studies could leverage InSAR datasets to enhance ground ice mapping within periglacial landforms by integrating knowledge of sediment grain size and soil moisture conditions."*

L. 400–401: This statement could be reconsidered to align with established observations. Thaw settlement typically stagnates when the thaw front passes through zones with minimal ice lenses (segregation ice) and intensifies when progressing through ice-rich layers.

Yes, we agree and discuss the theory you mention as first explanation for stagnation in thaw settlement in L. 350-355. However, as we describe in that paragraph, the stagnation seems to first occur when the thaw front reaches excess ice rich layers in our data. We therefore discuss that there might be an initial stagnation as the thaw front reaches ice layers, due to latent heat effects and slow meltwater drainage.

This interpretation from L. 350-355 is the basis for the statement in L. 400-401 and we therefore have left this sentence unchanged.

L. 407-409: This sentence could benefit from clarification. Could you elaborate on what is meant by "increase the influence on"? If you are referring to processes at the interface between the active layer and uppermost permafrost, I suggest adopting the concept of the "Transient Layer" as described by Shur et al. (2005).

By "increase the influence of melting excess ice," we meant that in exceptionally warm years, a deeper active layer thickness can cause thaw penetration into the transient layer or uppermost permafrost. Since both the transient layer and uppermost permafrost are often rich in excess ice (Shur et al., 2005), warmer summers can result in larger subsidence signals due to excess ice melt. However, we have restructured and updated Section 4.2 based on your and the other reviewers' comments. As a result, this specific sentence (L. 407-409) was removed, since we already discuss the late-season subsidence signal from excess ice melt in Section 4.1 (L. 370-371; L. 398-400).

To acknowledge the transient layer, we have now included it in L. 357:

*"The ground surface stabilization could have been caused by the thaw front reaching the ice-rich transient layer or uppermost permafrost (Shur et al., 2005)."*

L. 412–413: "Negative" should be replaced with "positive"? It seems to be a contradiction with the subsequent sentence. Please revise for consistency.

Thank you. Liu et al. (2012) do indeed assume a positive relationship, whilst our data indicates a negative relationship between pore ice melt and ALT. We have also realized that this section (4.2) was overall difficult to follow and have therefore updated the section by restructuring and adapting it based on your and the other reviewers' comments. The updated section 4.2 does not alter any of the main conclusions. The updated section 4.2 is now:

*"Previously, InSAR subsidence has been used for ALT inversion by assuming a positive relationship between InSAR subsidence observations of pore ice melt and ALT, with no consideration of excess ice contributions (Liu et al., 2012; Schaefer et al., 2015; Jia et al., 2017; Wang et al., 2018; Peng et al., 2023; Scheer et al., 2023). Some of these studies show good alignment between the InSAR-ALT and ALT from in-situ field measurements (Schaefer et al., 2015). However, recent findings show that the positive correlation between InSAR subsidence*

*observations of pore ice melt and ALT used as basis for the inversion is not valid in drier regions, where the correlation is negative (Chang et al., 2024).*

*We also find a negative correlation between ALT and expected pore ice subsidence (Fig. 4A), indicating that larger ALT can occur at sites with less pore ice. In addition, our results suggest that traditional inversion models are not universal and might not apply in areas with complex ground stratigraphy. Adventdalen is a periglacial valley which has a very diverse geomorphology with a variety of landforms, some of which have abundant excess ice in the active layer, as for example observed in outer alluvial fans (Fig. 6).*

*Our results indicate that the surface subsidence signal can be dominated by excess ice melt and drainage instead of pore ice melt, which complicates inversions from thaw subsidence to ALT. Further, the correlation between ALT and the InSAR subsidence was close to random in our results. Considering only pore ice melt to explain the observed subsidence in our study area would cause large errors, since the main subsidence contribution is from excess ice (Fig. 6).*

*In alignment with our observations (Fig. 7a), Antonova et al. (2018) also observed a poor match between ALT and subsidence.* They compared *in-situ* ALT measurements to in-situ surface subsidence from reference rods anchored in Yedoma permafrost in Siberia and found a weak positive correlation, with a low coefficient of determination. Further, when comparing the measured in-situ subsidence to the expected subsidence predicted by a pore ice melt model based on soil moisture measurements, they found a moderate alignment but noted significant outliers where the model underestimated the measured subsidence. Very large in-situ subsidence was for example measured in drained lake basins, yet not consistently.

Such outliers *in drained lake basins* have also been reported in InSAR-ALT estimates. Schaefer et al. (2015) reported outliers in *InSAR-ALT* estimates over drained lake basins at Barrow, Alaska, where high subsidence led to a large *InSAR-ALT* under the assumption of only pore ice melt contributing to the subsidence. Similar patterns of high subsidence in such landforms were also reported by Liu et al. (2014) on the Alaskan North Slope near Prudhoe Bay, by Strozzi et al. (2018) at *Teshekpuk* lake, Alaska, and by Bartsch et al. (2019) in central Yamal, Russia. Our results *suggest that the mismatch between InSAR-ALT and in-situ ALT is likely due to the omission of excess ice*, which has been previously detected in these landforms (Jorgenson and Shur, 2007; Bockheim and Hinkel, 2012; Kanevskiy et al., 2013).

*Overall,* our study indicates that the contribution from excess ice should not be neglected in models utilizing InSAR time series for active layer characterization, and that simple active layer models relying on constant pore ice contents are oversimplistic in periglacial environments like Adventdalen, Svalbard. *Future work should investigate integrating InSAR time series with numerical models that simulate ice segregation processes and excess ice content (e.g., Aga et al., 2023). This could help constrain model parameters or improve process representation by leveraging InSAR-derived surface deformation patterns, potentially through data assimilation techniques (e.g., Aalstad et al., 2018)."*

L. 416–418, 425–428: The ALT inversion described in these references is based on an incorrect frost heave theory that attributes ground movement to water volume expansion upon freezing rather than to water redistribution caused by ice segregation. Consider revising these interpretations to align with established frost heave mechanisms.

We acknowledge that these studies omit classic frost heave mechanisms, particularly the role of excess ice. The results of our study support your point. However, we believe it remains important to discuss these other studies, as they, in some cases, show a reasonable fit between InSAR ALT inversion and in-situ ALT measurements, despite notable outliers. As visible in our answer to your above comment regarding L. 412-413, we have strongly adapted this section, including L. 416-418 and L. 425-428. We point out our findings and present other recent observations, which indicate that an ALT inversion from a simple pore ice model is oversimplistic.

L. 415: The alignment between in-situ ALT probing and temperature measurements is not entirely clear. Could you elaborate on what this alignment reveals and how it connects to the following part of the sentence?

In this sentence we are discussing that some previous studies showed a good alignment between the InSAR-ALT based on Liu et al. (2012) and ALT from in-situ field probing and temperature measurements. We believe this is important to mention and have therefore kept this information also in the updated section 4.2 (see above answer regarding L. 412-413), but improved the clarity:

> *"Some of these studies show good alignment between the InSAR-ALT and ALT from in-situ field measurements (Schaefer et al., 2015)."*

L. 428: Correct "Teshepuk" to "Teshekpuk."

Corrected.

L. 432: I am very pleased with the conclusion drawn from your InSAR and in-situ investigations. Your work goes beyond merely addressing the oversimplification of AL-subsidence models; it highlights a critical oversight—the neglect of classic frost heave studies and the role of excess ice in such models. This omission in the previous works was surprising for me.

Thank you! We really appreciate your thorough review, which has improved our manuscript. We hope our answers could clarify your comments.

**References**

• Taber, S. (1930). The mechanics of frost heaving. J. Geol., 38, 303–317.

• Bai, R. et al. (2020). Investigation on frost heave of saturated–unsaturated soils. Acta Geotechnica, 15(11), 3295–3306. https://doi.org/10.1007/s11440-020-00952-6

• Shur, Y., Hinkel, K. M., & Nelson, F. E. (2005). The Transient Layer: Implications for Geocryology and Climate-Change Science. Permafrost and Periglacial Processes, 16, 5–17.

**References:**

Chang, T., Yi, Y., Jiang, H., Li, R., Lu, P., Liu, L., Wang, L., Zhao, L., Zwieback, S., and Zhao, J.: Unraveling the non-linear relationship between seasonal deformation and permafrost active layer thickness, npj Clim Atmos Sci, 7, 1–11, https://doi.org/10.1038/s41612-024-00866-0, 2024.

Derk, L. and Unold, F.: Effect of temperature gradients on water migration, frost heave and thaw-settlement of a clay during freezing-thaw process, Experimental Heat Transfer, 36, 585–596, https://doi.org/10.1080/08916152.2022.2062069, 2023.

Dirksen, C. and Miller, R. D.: Closed-System Freezing of Unsaturated Soil, Soil Science Soc of Amer J, 30, 168–173, https://doi.org/10.2136/sssaj1966.03615995003000020010x, 1966.

Dumais, S. and Konrad, J.-M.: Framework for thaw consolidation of fine-grained soils, Can. Geotech. J., 61, 931–944, https://doi.org/10.1139/cgj-2022-0502, 2024.

Everdingen, R. O. van: Multi-language glossary of permafrost and related ground- ice terms, International Permafrost Association, 1998.

French, H. M.: Cold-Climate Weathering, in: The Periglacial Environment, John Wiley & Sons, Ltd, 47–82, https://doi.org/10.1002/9781118684931.ch4, 2007a.

Fu, Z., Wu, Q., Zhang, W., He, H., and Wang, L.: Water Migration and Segregated Ice Formation in Frozen Ground: Current Advances and Future Perspectives, Front. Earth Sci., 10, https://doi.org/10.3389/feart.2022.826961, 2022.

Härtel, S. and Christiansen, H. H.: Geomorphological and Cryological map of Adventdalen, Svalbard, PANGEA, https://doi.pangaea.de/10.1594/PANGAEA.833048, 2014.

Rempel, A. W.: Formation of ice lenses and frost heave, Journal of Geophysical Research: Earth Surface, 112, https://doi.org/10.1029/2006JF000525, 2007.

Rouyet, L., Lauknes, T. R., Christiansen, H. H., Strand, S. M., and Larsen, Y.: Seasonal dynamics of a permafrost landscape, Adventdalen, Svalbard, investigated by InSAR, Remote Sensing of Environment, 231, 111236, https://doi.org/10.1016/j.rse.2019.111236, 2019.

---

## Author Comment (AC2)

**Answers to Reviewer 2**

**In the following, we answer (shown in blue) the copy-pasted comments from the reviewers (shown in black). Changes to the manuscript are shown in italics and the line numbers refer to the original preprint.**

The submitted article by Wendt et al. primarily compares seasonal ground surface displacement using InSAR remote sensing for 2023 to expected subsidence derived from ice contents from core drilling at 12 sites in Adventdalen, Svalbard. They establish a reasonable correlation between the InSAR-derived subsidence and that expected from the ground ice content in the cores, which includes determinations for pore ice, excess ice, and water drained upon the melt of excess ice. The authors determine that excess ice melt is the key contributor to observed subsidence at many of the sites, with pore ice typically being of secondary importance. The authors further demonstrate that without detailed knowledge of excess ice conditions, active layer thickness cannot be reliably estimated from InSAR in ice-rich terrain.

The strength of this paper lies in the fact that the InSAR subsidence trends can be partly, and fairly strongly, supported by the in situ ground ice determinations from immediately before the remote sensing record, which are commonly not available in similar remote sensing studies. The sampling scheme was well thought out and captured a significant range in ground ice conditions due to the selection of sites from different landforms and substrate conditions. Overall, I recommend this paper for publication however I have many minor comments and a couple more substantive ones that, if addressed, I think will strengthen the manuscript.

We would like to thank the reviewer for the detailed comments! These have been very useful and are addressed point by point below.

Based on feedback from both reviewers, we have added additional supplementary figures, which are currently numbered sequentially after the previously existing ones. In the revised manuscript and supplement, we will update the numbering and order of the supplementary figures to maintain a logical sequence when referring to them in the revised manuscript.

**Main comments/suggestions:**

**Ice wedges**
The ground resolution is stated as 18.2 x 28.2 m. Some of the sites include ice wedge polygons. In years of very deep thaw, presumably thaw would extent into the tops of ice wedges, and this could materially contribute to subsidence (e.g., https://doi.org/10.1002/ppp.2113). Based on the size of the polygons on Svalbard, I assume some pixels that included a core sample may have also included an ice wedge trough (or more than one). If this is the case, it should be discussed. Could this help explain why the 2023 InSAR derived subsidence is commonly higher than expected subsidence (Figure 6)?

This is a good point. Core E8 is located in an ice wedge polygon, and was extracted from its centre, while the InSAR pixel also covers the ice wedge troughs. Core E2 is also located in a polygon. We do not see a clear pattern of higher 2023 InSAR-derived subsidence compared to the expected subsidence and have therefore not discussed this further for Figure 6. Instead, we are now discussing this as part of the limitations of our study (section 4.3, line 461):

> *"Further, the site E8 is located within a low-centre ice-wedge polygon. Whilst the core was extracted at the polygon centre, the InSAR pixel is large enough to include thaw subsidence effects from the ice wedge troughs (Short and Fraser, 2023). Since the ice wedge tops in adjacent polygons have been observed to be located just below the active layer (O'Neill et al., 2025), the late-season subsidence observed at this site might partly reflect this (Burn et al., 2021). Another site, E2, also displays polygonal features indicative of ice wedges. Yet, these sites do not present as outliers in our analysis (Fig. 6)."*

**Date of snowmelt**

The InSAR record includes scenes following the melt of snowpack at the ADV met station. Did you examine whether snow had melted by this date in ice wedge troughs (or more generally in different topographic settings at different sites)? I presume snow may have persisted later, particularly in deeper troughs, as the snow depths are greater there. I observed this when I was on Svalbard. If this is likely to have occurred also in 2023, you may wish to consider what effect this may have had on the InSAR results at sites with ice wedges, or other settings where deeper snow could have accumulated, and include it in the discussion of limitations.

We agree that there is spatial variability in snow melt-out dates across the study area. We have reviewed the available Sentinel-2 imagery during this period and can confirm that snow cover was still present in topographic depressions, but all except one coring site were snow-free or had mixed pixels at the start of the InSAR time series. We have included this point as a limitation in section 4.3, line 462:

> *"The InSAR time series starts three days after the snow melt-out date at the Adventdalen meteorological station. Sentinel-2 imagery confirms that all sites except S5 were snow-free or had mixed pixels at the start of the InSAR time series, suggesting minimal impact of snow cover. However, initial subsidence just after snowmelt may not be fully captured at some sites, which could affect comparability. At S5, InSAR subsidence remained negligible until after local snowmelt, which occurred around 7 June 2023."*

**Thaw penetration and subsidence rates**

The role of pore ice in the nature of the subsidence curves over the summer could be better presented and discussed in relation to established theory and observations. During the thawing season, some of the sites follow a characteristic exponential decay curve in subsidence in layers where excess ice is not present. This generally follows the Stefan equation that described expected progression of active layer thawing. The pattern has been examined in relation to subsidence previously, for example in this paper that you cite in your discussion: https://tc.copernicus.org/articles/14/1437/2020/, and in other applications involving permafrost thaw. So, when the authors indicate that pore ice contributes to subsidence in a more "continuous

manner" l. 346, I don't think this is the best way to describe it. Though it is continuous, the rate is not. Furthermore, indicating that Schuh et al. 2017 confirmed the inverse relationship with ice content, while not inaccurate, is perhaps not the best option to support the relation observed. An inverse relation exists in the absence of excess ice, and governing equations that relate the thaw rate to the square root of time significantly predate the cited study. So, I suggest familiarization with the Stefan equation and the expected exponential decline of thaw progression with time and edits to associated text, and reference to pertinent literature.

Thank you for pointing this out. Our intent with saying "pore ice contributes in a more continuous secondary manner" was based on the fact that pore ice is less likely to appear as heterogenous layers with highly variable ice content compared to excess ice, and thereby does not cause step-wise subsidence signals. Also considering the feedback from reviewer 1, we have updated the discussion in l. 346 to:

> "We observed seasonal variations in the InSAR subsidence patterns, which especially align with the distribution of excess ice in the active layer. *Pore ice also contributes to the subsidence, but in a secondary manner, which more closely aligns with the thaw progression predicted by the Stefan equation (e.g. Fig. 5b: site E3).*"

We have also updated the reference for the inverse relationship between ice content and thaw progression to French (2007a).

**Figures**

Line 242 indicates that "Due to the dominant contribution of excess ice melt and drainage to the total subsidence…". However, this isn't presented or established until Figure 6, so the statement is confusing to the reader because this result hasn't yet been shown. Figure 6 showing this partition should come earlier. I understand this would entail showing the InSAR max earlier than the InSAR section, which is not ideal, but I think it is perhaps better overall because at least then the readers will be familiar with the expected subsidence in section 3.1.

We agree that the statement in its current form is confusing, as it presents information which first makes full sense when Figure 6 is shown. However, instead of moving Figure 6, we reformulated the statement in line 242 to:

> "The comparison between the measured in-situ ALT and the expected subsidence from pore ice melt revealed a strong negative correlation ($R^2 = 0.71$, Pearson's r = -0.84, Fig. 4A). *Conversely, there* was no correlation between ALT and the expected subsidence from only excess ice melt and drainage (Fig. 4B). *Overall, there was a poor correlation between the ALT and the total expected subsidence ($R^2 = 0.03$, Fig. 4C). This result suggests that the excess ice (not correlated with ALT, Fig. 4B) has a more significant contribution to the total expected subsidence than the pore ice (correlated with ALT, Fig. 4A). Results from Section 3.2 confirm this hypothesis.*"

It would be useful to have a figure showing photos of some of the site types: e.g., an Eolian one in ice wedge polygon fields, and alluvial example, slope/solifluction example, etc.

We have created an additional figure for the supplement (see below, Fig. S9), which shows photos from each site.

[Figure]

*Figure S9: Pictures of all coring sites from September 2023, taken with a drone approx. 20 m above ground (width of picture = approx. 30-40 m). The respective coring site is in the centre of each image.*

**Minor comments/suggestions:**

Line 16: change "allowing to estimate" to "allowing estimation of"
Updated.

Line 18: delete "thickness" after "active layer".
Updated.

Line 31. I presume here you mean increases in ALT are "largely influenced by the presence of ground ice" but the link is not explicit nor explained. Suggest restructuring this sentence.
We have removed the second half of the sentence, since the influence of ground ice on the ground thermal regime is explained in the following sentences. Additionally, we have added a reference to the GCOS ECV parameters, of which one is ALT:

"An increase in the active layer thickness (ALT) serves as a key indicator of permafrost degradation *(GCOS, 2022)."*

Line 34-35 "Long-term ground ice loss is associated…." This sentence should be supported by appropriate reference(s). This recent one covers the topics described: https://doi.org/10.1002/ppp.2261
Thanks, reference added.

Line 37. Consider indicating specifically which traditional methods you mean (e.g., probing, thaw tubes, dGPS surveys, etc).
We have updated the sentence to:

> "Traditional methods for monitoring ALT and mapping ground ice *(e.g. thaw depth probing, temperature monitoring in boreholes, drilling and geomorphological surveys, thaw tubes)* typically rely on labour-intensive, time-consuming in-situ surveys."

Line 39. Consider examining use of "utilized" throughout the text and replace with "used", which is more concise and generally has the same meaning.

Updated to "used" throughout the text.

Line 43. "stronger consolidation" suggest changing this to "larger magnitude thaw subsidence".

We have updated this sentence to:

> "Excess ice melt can cause a *larger thaw subsidence magnitude*, since the resulting meltwater exceeds the soil pore space and may drain away (Morgenstern and Nixon, 1971)."

Line 44. "likely drains" suggest changing to "may drain away."

Updated.

Figure 1. The stream and lake (reservoir) colour is the same as InSAR heave; suggest changing all waterbodies to a colour not in the InSAR legend. The inset map of Svalbard is very hard to discern, and the contrast between land and water is poor. Consider enlarging and colour changes.

We modified Figure 1 (see below) to account for your comments.

[Figure]

*Figure 1: (a) The Adventdalen study area with the location of the coring sites and their label names. The background is the maximum InSAR seasonal displacement of 2023. Subsidence is shown with negative values (red) and heave with positive values (blue). Note that the color scale is saturated for visualization. (b) Simplified geomorphological map of the study area with the main sediment deposits (Rouyet et al., 2019; modified from Härtel and Christiansen, 2014). Background: hillshade of a digital elevation model (Norwegian Polar Institute, 2014b). Coordinate System: WGS 1984 UTM 33N.*

Line 105. Add "anticipated" before "InSAR subsidence magnitudes" since 2023 magnitudes were not

known when sites were selected.

Added.

Line 111. Clarification on sampling – "soil moisture conditions were considered to include dry and wet locations". Explain specifically how soil moisture conditions were considered. I presume there were not soil moisture instruments at each site, and that this was done based on some visual or field interpretation?

Indeed, the soil moisture conditions were assessed based on the NDWI remote sensing index applied to Sentinel-2 imagery from summer 2022 (Gao, 1996). We have extended the sentence to:

"Further, soil moisture conditions were considered to include dry and wet locations *based on the NDWI (Gao, 1996) remote sensing index of Sentinel-2 imagery from summer 2022.*"

Line 117. "0.5 m core" is ambiguous, indicate this is the core length.

Updated the sentence to:

"The drilling was conducted using a STIHLTM BT 121 Earth Auger *equipped with 0.5 m long core barrels."*.

Line 119. "freezing container" is unclear. Do you mean a cooler? Or something that actively freezes contents?

Yes, we do mean a container with an active cooling system. We have updated the sentence to:

"Cores were retrieved in 5–30 cm sections and were immediately packed, air-sealed, and stored at the end of each field day *in a container with an active freezing system."*

Line 127. Add "area" after "surrounding".

Added.

Table 1. Header for column 5 does not indicate that the information at the top of the cell is the ALT measurement date. Row E8 – "Drill location in center" of what (ice wedge polygon)?

Thank you! We have updated the header for column 5. We also added in row E8 and E2 *"Drill location in centre of ice wedge polygon.".*

Line 139. What classification was used to classify cryostratigraphy? A reference should be provided.

We used the classification from French and Shur (2010) and have updated this sentence to:

"The intact subsections were scraped, the cryostratigraphy classified *based on French and Shur (2010),* and visual ice content and sediment type described."

Line 151. "the factor 1.09 represents the density of ice relative to water". The equation deals with volumes, so it is better to say that the 1.09 is to "estimate the equivalent volume of ice" from the water volume as Kokelj and Burn did.

We have updated this sentence part accordingly to:

"*and the factor 1.09 is to estimate the equivalent volume of ice from the water volume."*

Line 156. This should be 9% shouldn't it? This is why the factor in Eq. 2 is 1.09. This would also affect your derivation of Eq. 4. You have 0.92 in Eq. 4 but this should be 0.912 I think, so rounded to 0.91. So, you will likely have to redo your calculations though they won't differ much. I think the

confusion/error lies in the fact that the percent difference is 9.2% (e.g., see percent difference equation at https://www.calculatorsoup.com/calculators/algebra/percent-difference-calculator.php). This depends on if you are converting from a water volume to an ice volume, or vice versa. From water to ice, the volume change is indeed +9.1 % relative to the initial water volume. From ice to water, the volume change is -8.3 % relative to the initial ice volume. These calculations are based on a water density of 1000 kg/m$^3$ and an ice density of 917 kg/m$^3$. Since Eq. 4 considers the transition from an initial ice volume to a water volume, the volume reduction is ~8 %. We therefore did not adjust our calculations.

Line 164. Change "which" to "that".
Updated.

Line 176. Can you clarify to the reader whether "temporal baseline" is synonymous with the return frequency of the satellite?
We have updated this section to:

> *"For summer 2023, only Sentinel-1A imagery was available, which has a revisit period of 12 days. Therefore, the minimum temporal baseline for constructing interferograms was 12 days.* To mitigate temporal decorrelation and phase ambiguities from strong subsidence in the exceptionally warm summer 2023, a maximum temporal baseline of 24 days was used."

Line 227. Indicate it's expected for two-sided freezing specifically.
Thanks, the sentence is updated to:

> "This pattern is consistent with the expected distribution of ground ice from *two-sided freezing* (French, 2007).".

Line 295. Delete "coring" and "located", these words are not required. Also, suggest changing "rather" to "mostly" in second sentence.
We changed "rather" to "mostly".
We did not remove "coring" in "A1 coring site", since we use the term coring site throughout the manuscript when we refer to our sites.

Line 316. Add "sand" after "dry".
Added.

Line 317. Add "early in the thawing season" after "quickly"
Added.

Figure 8. "Grain type" figure title should perhaps be changed to "Material type" because organic is not a grain type, and neither is ice lens, or disturbed.
We have updated this in Fig. 8, S5, S6, and S7.

Line 341. Add "from the active layer and upper permafrost" after "in situ-ice contents".
Thanks, added.

Line 357. Remove "excess" because ice-rich permafrost, by definition, includes excess ice. Check this throughout.

We have updated "excess ice-rich" to "ice-rich" throughout the manuscript. We only kept "excess ice-rich" when referring to results of the study of Zwieback and Meyer (2021), who used this term in their work.

Line 365. I think it would be good to give a few examples of sites from Figure 6 where it dominates (e.g., the A sites).

We are now referring in this line to Figure 6:

"The expected subsidence from pore ice melt lies within a plausible range, yet our results indicate that excess ice melt and meltwater drainage can significantly dominate the expected subsidence signal *(Fig. 6, A1, A9, A12).*".

Line 371. This part could be strengthened by giving examples of the magnitudes/proportions accounted for by excess ice in the late thawing season.

We cannot provide exact numbers for this, since we do not have thaw front progression data for the different sites. We therefore only refer to the comparison figures per coring site, which display the InSAR subsidence time series and the respective in-situ ground ice content measurements:

"The findings of Zwieback and Meyer (2021) align with our results, which indicate that excess ice can be the major contribution to the InSAR late-season subsidence signal *(Fig. 8, S5-7)."*

Line 393. "drainage variations can control the presence or absence of excess ice". While I don't disagree, because fundamentally moisture is required for ice formation, based on detailed coring I conducted in the eolian sediments (a GSC Open File is now in press), excess ice was mainly controlled by grain size of the eolian materials, regardless of present-day moisture conditions in the polygons (the polygon with standing water and wet active layer had on average half the excess ice content in the top 1 m of permafrost). Siltier layers, which imply slower rates of loess aggradation and different climatic, eolian source conditions, and likely microtopography in different time periods, were associated with higher ice contents. You have not measured "lateral drainage" (l. 395) in this study, though you may have observed it at the surface. Also, it is hard to know whether drainage conditions at the surface today reflect those when the ground ice aggraded in the past, as the syngenetic polygons fields are dynamic. Therefore, you cannot confidently say that the drainage conditions are controlling the ice contents at much depth beyond the current permafrost table; this text should be modified.

Thank you for sharing your observations. In the meantime, your GSC Open File was published, and we have included it. We have updated this section to:

*"Our data exemplify that, even within the same type of sediment deposit excess ice presence can vary in the active layer and uppermost permafrost. For instance, eolian fine-grained loess terraces show significant variability: some have very low ground ice contents and lack excess ice (e.g., coring site E10), while others have large ground ice contents, particularly in the lower active layer (e.g., coring sites E2, E8) (Fig. 8, S5). This is likely related to drainage and grain size variations (O'Neill et al. 2025), as well as the site-specific formation history of the sediments (Gilbert et al. 2018)."*

Line 412 last word: Should be "basis".
Updated.

Line 435. This sentence suggests that InSAR time series could be used in conjunction with models incorporating ice segregation processes/excess ice content. The reader is left confused what the objective of such an exercise is. Is it to use the excess ice content from the model to validate an InSAR signal? If so, this would surely not be appropriate given that such models cannot accurately capture conditions that lead to excess ice formation over hundreds or thousands of years, and thus cannot produce accurate estimations of ground ice at the site scale. This is discussed in, e.g.,: https://agupubs.onlinelibrary.wiley.com/doi/full/10.1029/2023JF007262. If this is not the intent, then can you please clarify specifically what you mean in terms of "combining" InSAR with such models?
We updated this sentence to:

> "Future work should investigate integrating InSAR time series with numerical models that simulate ice segregation processes and excess ice content (e.g., Aga et al., 2023). This could help constrain model parameters or improve process representation by leveraging InSAR-derived surface deformation patterns, potentially through data assimilation techniques (e.g., Aalstad et al., 2018)."

Line 444. At this time of the year, the whole active layer is frozen, so the meaning of this sentence is unclear. Please clarify what you mean in terms of where/how water is moving at this time in relation to the anticipated ground temperature gradient(s) from the ground surface to the upper permafrost at the time of drilling.
We did not mean that ground ice contents would change at the time of drilling, but rather that at the beginning of the thawing season water could infiltrate into the ground and refreeze, thereby changing the in-situ ground ice contents. Such infiltration of meltwater and refreezing has been observed before and our cores would in such a case underestimate the amount of in-situ ground ice. We have updated this sentence to:

> "The cores were collected at the end of the freezing period, *yet water infiltration at the thaw season onset into the frozen part of the sediment column could have caused aggradational ice growth after core collection* (e.g. Mackay, 1983, *Scherler et al. 2010*)."

Line 462. Remove "excess".
Removed.

Line 463. Why might probing be less precise than borehole measurements? This is highly dependent on the spacing of thermistors, and the material being probed in. This should either be explained further or amended.
We have updated this sentence to:

> "In this study manual probing was employed, which allows a more widespread coverage of the pixel, *yet is dependent on rather fine-grained ground conditions. Additional borehole temperature measurements could have aided in determining the ALT and provided thaw progression measurements throughout the thawing season, which would have been valuable for comparison against the InSAR subsidence progression.*"

**References:**

Burn, C. R., Lewkowicz, A. G., and Wilson, M. A.: Long-term field measurements of climate-induced thaw subsidence above ice wedges on hillslopes, western Arctic Canada, Permafrost and Periglacial Processes, 32, 261–276, https://doi.org/10.1002/ppp.2113, 2021.

French, H. M.: Cold-Climate Weathering, in: The Periglacial Environment, John Wiley & Sons, Ltd, 47–82, https://doi.org/10.1002/9781118684931.ch4, 2007a.

French, H. and Shur, Y.: The principles of cryostratigraphy, Earth-Science Reviews, 101, 190–206, https://doi.org/10.1016/j.earscirev.2010.04.002, 2010.

Gao, B.: NDWI—A normalized difference water index for remote sensing of vegetation liquid water from space, Remote Sensing of Environment, 58, 257–266, https://doi.org/10.1016/S0034-4257(96)00067-3, 1996.

GCOS. The 2022 GCOS implementation plan. Technical Report GCOS-244, World Meteorological Organization, Geneva, Switzerland, 2022.

Gilbert, G. L., O'Neill, H. B., Nemec, W., Thiel, C., Christiansen, H. H., and Buylaert, J.-P.: Late Quaternary sedimentation and permafrost development in a Svalbard fjord-valley, Norwegian high Arctic, Sedimentology, 65, 2531–2558, https://doi.org/10.1111/sed.12476, 2018.

O'Neill, H. B., Gilbert, G. L., and Christiansen, H. H.: Site-scale variation in ground-ice content and physical properties of loess in permafrost, Svalbard, High Arctic, https://doi.org/10.4095/pfmq507fg6, 2025.

Scherler, M., Hauck, C., Hoelzle, M., Stähli, M., and Völksch, I.: Meltwater infiltration into the frozen active layer at an alpine permafrost site, Permafrost and Periglacial Processes, 21, 325–334, https://doi.org/10.1002/ppp.694, 2010.

Short, N. H. and Fraser, R. H.: Comparison of RADARSAT-2 and Sentinel-1 DInSAR displacements over upland ice-wedge polygonal terrain, Banks Island, Northwest Territories, Canada, https://doi.org/10.4095/331683, 2023.

---

## Referee Report (RR1)

Comments to Revised 1

Thank you for including the core cross sections, which have helped answer many of my questions. This manuscript was significantly improved by addressing the reviewers' questions, and revisions were made in accordance with their comments. I consider this work an important contribution to the remote sensing of frozen ground behavior, with field validation, which is rare.

However, I must still address an essential discussion for authors to consider in their future research, as follows. I defer to the authors' decision whether to respond to my comments and incorporate them into their manuscript or discuss them in their future works.

The key concern points of my major comments 1-3 were the contribution of volumetric loss to pore ice melt. Please note that I am in total agreement with the 100% contribution of excess ice melt to seasonal subsidence. Although we are on the same page in the fact that the primary contribution to the seasonal thaw settlement is due to the loss of excess ice (in other words, the primary cause of the frost heave is ice segregation/ice lens formation due to soil water redistribution), my concern is that the full contribution of 8 % pore ice to the thaw settlement must be a significant overestimate (I am aware that authors excluded very dry portions from this discussion).

For example, my major comment2 "···20% to total subsidence (as shown in Fig. 6) … "(AC2 in your response) is stating about the contribution % in the total subsidence, for example, about 20 % (20mm PIC contribution against 100mm total subsidence). for A1, about 50 % for E8, or 100 % for E10.

[Figure]

**Figure 6:** Contribution of pore ice melt, excess ice melt, and excess ice meltwater drainage to the total expected subsidence for each coring site. The maximum InSAR subsidence in 2023 is displayed as grey diamonds. The x-label denotes the site names (sediment deposit type and unique core number, see also Table 1).

According to the authors' estimation model of thaw subsidence from core analysis, most of the upper AL at their sites frost-heaved 100% due to water phase expansion, as they contained no excess ice (see the examples of A1/E8/E10 below).

Although I agree that this happens in a close-system of fully saturated soil, it is unlikely in unsaturated soils or even in fully saturated soil in an open-system (in the case of our argument, the freezing front goes top down, and the water expansion pressure can escape to the air space in the unsaturated zone beneath (or 3-dimensionally/laterally). In your study sites, E3 may have had a condition of waterlogged (saturated) bog, where 9 % volume expansion of pore water could fully contribute to the frost heave amount. However, other sites seem to have unsaturated conditions, where I cannot imagine a closed system in the AL.

[Figure]

[Figure]

The no-EIC zone in AL is fully saturated or unsaturated (not over-saturated). Judging from the soil texture and VIC of the above three examples (A1, E8, and E10), the AL of most locations appears to be largely unsaturated (I am aware that the authors have already excluded highly unsaturated and disturbed soil layers from the subsidence estimation). Even though the AL was saturated just before freezing and the soil water was redistributed, producing frost heave, the middle layers of the frozen AL tend to have a desiccated zone due to cryosuction and water redistribution. And the pore waters in those unsaturated zones should not be regarded as contribution sources to the subsidence.

My understanding and knowledge from previous studies on frost heave were that there was a negligible contribution from pore-water expansion upon freezing to the amount of frost heave in the most natural ground surface layers.

Therefore, accounting for all PIC in the AL to contribute to frost heave/thaw settlement can be a significant overestimate in this study, although the volume reduction is only 8% of them, and although the absolute subsidence calculated is relatively small. This argument may considerably influence the validation assessment between InSAR and expected subsidence, especially when the contribution of pore ice melt is significant enough.

---

## Author Response (AR2)

**Response to referee report 1 for: "InSAR sensitivity to active layer ground ice content in Adventdalen, Svalbard"**

**In the following, we answer (shown in blue) the comments from the referee (shown in black). Changes to the manuscript are shown in italics.**

Thank you for including the core cross sections, which have helped answer many of my questions. This manuscript was significantly improved by addressing the reviewers' questions, and revisions were made in accordance with their comments. I consider this work an important contribution to the remote sensing of frozen ground behavior, with field validation, which is rare.

However, I must still address an essential discussion for authors to consider in their future research, as follows. I defer to the authors' decision whether to respond to my comments and incorporate them into their manuscript or discuss them in their future works.

The key concern points of my major comments 1-3 were the contribution of volumetric loss to pore ice melt. Please note that I am in total agreement with the 100% contribution of excess ice melt to seasonal subsidence. Although we are on the same page in the fact that the primary contribution to the seasonal thaw settlement is due to the loss of excess ice (in other words, the primary cause of the frost heave is ice segregation/ice lens formation due to soil water redistribution), my concern is that the full contribution of 8 % pore ice to the thaw settlement must be a significant overestimate (I am aware that authors excluded very dry portions from this discussion).

For example, my major comment2 "...20% to total subsidence (as shown in Fig. 6) …

"(AC2 in your response) is stating about the contribution % in the total subsidence, for example, about 20 % (20mm PIC contribution against 100mm total subsidence). for A1, about 50 % for E8, or 100 % for E10.

[Figure]

**Figure 6:** Contribution of pore ice melt, excess ice melt, and excess ice meltwater drainage to the total expected subsidence for each coring site. The maximum InSAR subsidence in 2023 is displayed as grey diamonds. The x-label denotes the site names (sediment deposit type and unique core number, see also Table 1).

According to the authors' estimation model of thaw subsidence from core analysis, most of the upper AL at their sites frost-heaved 100% due to water phase expansion, as they contained no excess ice (see the examples of A1/E8/E10 below).

Although I agree that this happens in a close-system of fully saturated soil, it is unlikely in unsaturated soils or even in fully saturated soil in an open-system (in the case of our argument, the freezing front goes top down, and the water expansion pressure can escape to the air space in the unsaturated zone beneath (or 3-dimensionally/laterally). In your study sites, E3 may have had a condition of waterlogged (saturated) bog, where 9 % volume expansion of pore water could fully contribute to the frost heave amount. However, other sites seem to have unsaturated conditions, where I cannot imagine a closed system in the AL.

[Figure]

The no-EIC zone in AL is fully saturated or unsaturated (not over-saturated). Judging from the soil texture and VIC of the above three examples (A1, E8, and E10), the AL of most

locations appears to be largely unsaturated (I am aware that the authors have already excluded highly unsaturated and disturbed soil layers from the subsidence estimation). Even though the AL was saturated just before freezing and the soil water was redistributed, producing frost heave, the middle layers of the frozen AL tend to have a desiccated zone due to cryosuction and water redistribution. And the pore waters in those unsaturated zones should not be regarded as contribution sources to the subsidence.

My understanding and knowledge from previous studies on frost heave were that there was a negligible contribution from pore-water expansion upon freezing to the amount of frost heave in the most natural ground surface layers.

Therefore, accounting for all PIC in the AL to contribute to frost heave/thaw settlement can be a significant overestimate in this study, although the volume reduction is only 8% of them, and although the absolute subsidence calculated is relatively small. This argument may considerably influence the validation assessment between InSAR and expected subsidence, especially when the contribution of pore ice melt is significant enough.

We thank the reviewer for highlighting this uncertainty. We agree that the contribution of pore ice melt may be overestimated in partially unsaturated soils that act as open systems. In our analysis, visibly unsaturated or disturbed core sections were not included in the expected subsidence calculation to avoid this bias. The 8% volumetric contribution was only applied to intact core sections, which were classified as pore-ice saturated after visual examination. Nevertheless, we acknowledge that open-system behaviour cannot be ruled out where lateral or vertical escape of expansion pressure would reduce frost heave and subsequently lead to less pore-ice-melt-induced subsidence than the theoretical 8% volumetric reduction assumed here. We have clarified this in the discussion of the limitations and highlighted it as a priority for future field validation as follows (L. 517 in revised manuscript):

> "Lastly, pore ice melt from core sections that were retrieved intact and appeared saturated was considered to contribute to the expected subsidence in this study. However, open-system behaviour of the active layer could have allowed lateral or vertical escape and expansion pressure from phase change to dissipate, leading to less pore-ice-melt-induced subsidence than the theoretical 8% volumetric reduction. This may have caused an overestimation of the expected subsidence from pore ice melt and could have affected the validation assessment between InSAR and expected subsidence. Future research should aim to better quantify the role of pore ice in heave-subsidence soil mechanics under natural freezing and thawing conditions."